# Taming Stochastic Gradient Descent: Almost Sure Convergence and Saddle-Point Avoidance under $(L_0, L_1)$-Smoothness

**Vassilis Apidopoulos** [1] [2] **Iosif Lytras** [2] **Panayotis Mertikopoulos** [3] [2]

## Abstract

Many optimization problems in machine learning and data science—from deep neural networks to Bayesian inference and beyond—fall outside the standard Lipschitz smoothness framework that underpins the convergence theory of stochastic gradient descent (SGD). Motivated by this theory-practice disconnect, we examine the almost sure convergence of the trajectories of SGD in non-convex landscapes under a generalized $(L_0, L_1)$-smoothness condition which allows for gradients with superlinear growth (even exponential). We begin by proposing a taming scheme for SGD that achieves almost sure convergence under a generalized ABC-type condition on the gradient noise. Subsequently, to relax this requirement, we introduce a more flexible, dissipative taming scheme which converges almost surely under less restrictive moment bound conditions for the stochastic gradients entering the process. For both taming schemes, we show that the generated trajectories avoid strict saddle points (and/or manifolds thereof) with probability 1 so, generically, both methods only converge to local minimizers.

## 1 Introduction

**Background.** Contemporary machine learning models and architectures—from neural networks to dictionary learning and inverse problems—invariably boil down to solving a high-dimensional non-convex optimization problem of the form

$$\min_{x \in \mathbb{R}^d} f(x) \qquad \text{(Opt)}$$

for some differentiable loss function $f \colon \mathbb{R}^d \to \mathbb{R}$. When the problem's dimension is so high as to render exact gradient computations prohibitively expensive, the *de facto* method for solving (Opt) is the *stochastic gradient descent* (SGD) algorithm

$$x_{t+1} = x_t - \gamma_t \hat{g}_t \qquad \text{(SGD)}$$

where $\gamma_t$ is the method's step-size, and $\hat{g}_t$, $t = 1, 2, \ldots$, is a sequence of stochastic approximations of $\nabla f(x_t)$ that are cheaper to compute than exact gradients.

Historically, (SGD) was introduced by Robbins & Monro [68] and Kiefer & Wolfowitz [41] as a method for solving systems of nonlinear equations in the computationally starved backdrop of the 1950's. Still, despite the fact that (SGD) has been around for more than 75 years, it remains the method of choice for training massively large machine learning systems—from large language models, to recommender systems and transformers. As a result, the study of (SGD) has generated a vast corpus of literature in optimization, probability and statistics, for both convex and non-convex problems.

The first convergence result for (SGD) in a general, non-convex setting was obtained by Ljung [48, 49, 50], who showed that $x_t$ converges with probability 1 (w.p.1) to a set of critical points of $f$, provided that $\sup_t \|x_t\| < \infty$. Even though this boundedness caveat is often impossible to verify from first principles, it persisted in the stochastic approximation literature [5, 43] until the foundational work of Bertsekas & Tsitsiklis [8] who showed that $\nabla f(x_t) \to 0$ w.p.1 without resorting to any boundedness assumptions. Ever since these landmark results, there have been a number of refinements and extensions, including the almost sure avoidance of saddle points [6, 14, 20, 24, 33, 37, 59, 64], extensions to non-gradient systems [5, 12, 43], manifold optimization [11, 13], and many other settings.

**The role of smoothness and related work.** A shared baseline—and common underlying assumption—in these works is that the problem's objective satisfies a (global) Lipschitz smoothness condition of the form

$$f(x') \le f(x) + \langle \nabla f(x), x' - x \rangle + (L/2)\|x' - x\|^2 \quad \text{(LS)}$$

---

[1]Athens University of Economics & Business, Athens, Greece. [2]Archimedes, Athena Research Center, Athens, Greece. [3]Univ. Grenoble Alpes, CNRS, Inria, Grenoble INP, LIG, 38000 Grenoble, France. Correspondence to: Vassilis Apidopoulos <v.apidopoulos@athenarc.gr>.

*Proceedings of the 43$^{rd}$ International Conference on Machine Learning*, Seoul, South Korea. PMLR 306, 2026. Copyright 2026 by the author(s).

for some $L > 0$ and all $x, x' \in \mathcal{X}$. This assumption serves a dual purpose: first, it provides a local descent inequality which is essential for the iterative analysis of (SGD); second, it ensures that $f(x)$ grows at most quadratically—and $\|\nabla f(x)\|$ at most linearly—as $\|x\| \to \infty$. This "tame growth" requirement is crucial for establishing the convergence and other properties of (SGD), because the method's *stability analysis*—i.e., showing that $x_t$ remains bounded w.p.1—often hinges on a second-order Taylor expansion.

On the other hand, this smoothness condition breaks down irrevocably when the loss landscape of $f$ grows at a superquadratic rate—e.g., as in deep nets, where losses exhibit polynomial growth with a degree proportional to the number of layers in the network. In turn, this creates a fundamental tension between theory and practice, whereby a linchpin of the analysis of (SGD) fails to hold in some of the method's most impactful application domains.

The effort to mitigate this disconnect has led to a flurry of activity with the aim of replacing (LS) with a "generalized" variant that concurrently allows for superlinearly growing gradients while maintaining local control (in the form of a suitable descent inequality). One of the most widely used conditions of this type is the notion of $(L_0, L_1)$-*smoothness*, which, in its simplest incarnation, posits that

$$\|\nabla^2 f(x)\|_F \leq L_0 + L_1 \|\nabla f(x)\| \qquad \text{(L}_0\text{L}_1\text{)}$$

for suitable constants $L_0, L_1 > 0$ and all $x \in \mathbb{R}^d$.[1] This condition was introduced by Zhang et al. [83, 84] as a surrogate for a large class of differentiable functions that exhibit polynomial gradient growth (or even exponential). These functions do not satisfy the global version of (LS), so it is not possible to construct a suitable quadratic majorant at each point; however, thanks to (L$_0$L$_1$), it is still possible to construct a global model capturing the geometry of the objective function (and, in particular, its growth envelope).

In turn, this has led to the introduction of several gradient shaping mechanisms, mostly based on clipping [7, 62, 63] and normalization [31]; for a repesentative list of references, cf. [7, 28, 31, 63, 84] and Appendix A. These methods have been proven to be remarkably effective in dealing with $(L_0, L_1)$-smooth objectives: In the deterministic non-convex setting Zhang et al. [83, 84] derived a series of iteration complexity bounds for $\|\nabla f(x_t)\|$ and $(1/T) \sum_{t=1}^{T} \|\nabla f(x_t)\|$, see also [17, 42, 79]. In the convex case, Koloskova et al. [42] and Gorbunov et al. [29] derived a series of improvements for clipped gradient descent in the (strongly) convex case, which can be coupled further with acceleration techniques or Polyak step-sizes [29, 45, 79].

In the stochastic non-convex setting—our paper's main focus due to its connection to deep learning—most guarantees

---

[1]Here $\|A\|_F = \sqrt{\text{tr}(A^\top A)}$ denotes the Frobenius norm of $A$.

for clipped/normalized SGD concern "regret-type" bounds of the form $\mathbb{E}[\sum_{t=1}^{T} \|\nabla f(x_t)\|^2] = \mathcal{O}(\sqrt{T})$. Following the "random stopping" trick popularized by Ghadimi & Lan [25], bounds of this sort can produce a (randomized) output point $\bar{x}_T$ with $\mathbb{E}[\|\nabla f(\bar{x}_T)\|^2] = \mathcal{O}(\varepsilon)$ after $T = \mathcal{O}(1/\varepsilon^2)$ iterations on average, cf. [42, 45, 83] and references therein. Finally, complementing this analysis blueprint with stopping time arguments, [18, 23, 45] derived a series of high probability guarantees under generalized smoothness.

For further elements of related work, see Appendix A.

**Our contributions.**    All these results provide efficient and streamlined methods of obtaining a point with low gradient norm (on average or in high probability), but the long-run behavior of stochastic gradient methods in non-convex landscapes with superlinearly growing gradients remains opaque—for example, it is not even possible to tell if the produced point lies in the vicinity of a minimizer, a saddle point, or even a local maximum of $f$. In view of this, our point of departure can be summarized as follows:

*Is it possible to obtain almost sure convergence guarantees in non-convex optimization without global smoothness? And, if so, are saddle points avoided in the long run?*

Our approach is based on the observation that similar stability issues arise in the numerical analysis of stochastic differential equations when the drift of the stochastic differential equation (SDE) exhibits superlinear growth [35]. Concretely, standard numerical integration techniques break down when the drift of an SDE—corresponding to $\nabla f$ in our case—exhibits superlinear growth. A principled paradigm to restore stability in this context is the technique of *taming*, originally due to Hutzenthaler et al. [36] and Sabanis [71, 72], which consists of modifying the drift of the SDE under study in an asymptotically vanishing way, and then using exponential moment bound and/or isoperimetric techniques to control the resulting "tamed" trajectories.

Motivated by this observation, we introduce a taming scheme for SGD—which we call *tamed stochastic gradient descent* (TSGD)—and which can be seen as running (SGD) with an adaptive, position-dependent step-size. By tethering the scale of the taming scheme to the smoothness profile of $f$, we show that the trajectories of the method converge with probability 1 to the set of critical points of $f$, thereby extending the seminal result of Bertsekas & Tsitsiklis [8] to the $(L_0, L_1)$-smooth setting. In addition, we also propose a dissipative version of TSGD, which allows more flexibility in the choice of taming and step-size parameters, and which we also show converges w.p.1. Finally, by establishing a link between taming and the theory of stochastic approximation [5], we show that the proposed taming methods also avoid strict saddle points (a.s.), thus recovering and generalizing existing results for (SGD) in the Lipschitz case [6, 59].

We are not aware of any comparable results in the literature on generalized smoothness, and we find both particularly appealing for the applicability of taming to deep learning loss landscapes, where saddle points are ubiquitous.

## 2 Problem setup and preliminaries

We now proceed to describe and discuss in detail our standing assumptions for (Opt) and the oracle providing gradient information on $f$. Some of these requirements may be relaxed (depending on the precise setting), but we refrain from doing so to avoid unnecessary technicalities and to have the cleanest presentation of our analysis and results.

**2.1. Assumptions on the objective.** We begin with our assumptions for $f$, writing $\mathcal{X} \equiv \mathbb{R}^d$ for the domain of (Opt).

**Assumption 1.** The objective function $f \colon \mathcal{X} \to \mathbb{R}$ of (Opt) satisfies the following conditions:

(*a*) *$C^2$-smoothness:* $f$ is two times continuously differentiable on $\mathcal{X}$.

(*b*) *Gradient coercivity:*

$$\liminf_{\|x\| \to \infty} \frac{\langle \nabla f(x), x \rangle}{\|x\|^2} > 0 \qquad \text{(GC)}$$

that is, $\langle \nabla f(x), x \rangle = \Omega(\|x\|^2)$ for large $x$.

These requirements are fairly standard, the first for streamlining statements regarding the characterization of the critical points of $f$, and the second as a "sanity check" that guarantees the existence of solutions to (Opt) and disqualifies instances with near-critical behavior at infinity—e.g., as in the case of $f(x) = \log(1 + \|x\|^2)$. In particular, (GC) readily implies that

(*a*) $f$ is bounded from below, i.e.,

$$f_* := \inf_{x \in \mathcal{X}} f(x) > -\infty. \qquad \text{(1a)}$$

(*b*) The critical set

$$\operatorname{crit} f = \{x \in \mathcal{X} : \nabla f(x) = 0\} \qquad \text{(1b)}$$

of $f$ is nonempty and compact.

We will use both facts freely in the sequel.

Moving forward, as we discussed earlier, many problems of central interest to machine learning may exhibit *super-quadratic* growth, so, in terms of smoothness and growth requirements, we will assume that $f$ is $(L_0, L_1)$-smooth.

**Assumption 2.** $f$ satisfies ($L_0L_1$).

To connect with the literature, Assumption 2 with $f$ assumed $C^2$ is equivalent to the (1-symmetric) first-order condition

$$\|\nabla f(x') - \nabla f(x)\| \leq \big(L_0 + L_1 \sup_{u \in [x,x']} \|\nabla f(u)\|\big) \cdot \|x' - x\| \qquad \text{(2)}$$

for all $x, x' \in \mathcal{X}$, cf. [17, 29, 79, 83] and references therein. At the cost of a more complicated analysis and presentation, (2) can be used as a stand-in for objectives that are not $C^2$-smooth, but this will not be important for our purposes. By contrast, an immediate consequence of ($L_0L_1$) which will play a central role in our analysis is the expansion

$$\begin{aligned} f(x') &\leq f(x) + \langle \nabla f(x), x' - x \rangle \\ &\quad + \frac{L_0 + L_1 \|\nabla f(x)\|}{2} e^{L_1 \|x' - x\|} \cdot \|x' - x\|^2, \quad \text{(3)} \end{aligned}$$

which holds for all $x, x' \in \mathcal{X}$ [17, Prop. 1]. This variant of (LS) lends itself directly to a (local) descent lemma, so (3) is often taken as the starting point for the analysis of gradient methods in this context; for a representative selection of works on the topic, cf. [17, 29, 42, 78, 79, 83, 84] and references therein.

A key point worth noting here is that the Assumption 2 imposes a worst-case growth bound of the form $\|\nabla f(x)\| = e^{\mathcal{O}(\|x\|(1+o(1)))}$ (see Lemma B.1 and Remark 5 in Appendix B), showing in this way that $(L_0, L_1)$-smoothness allows for loss functions with at most *exponentially* growing gradients. This black-box upper bound on the growth of $\|\nabla f(x)\|$, issuing from $(L_0, L_1)$-smoothness, plays a central role on the design of the proposed methods (see Sections 3 & 4). An important example of this type is deep neural networks with continuously differentiable, unbounded activation functions—such as softplus, exponential and/or Gaussian error linear units, etc. In this case, the resulting loss function exhibits polynomial growth with a degree that scales linearly with the order of the number of layers $L$ in the network—typically as $\mathcal{O}(\|x\|^L)$ for hinge losses, and as $\mathcal{O}(\|x\|^{2L})$ under quadratic losses.

**2.2. Assumptions on the oracle.** To compute gradient information for $f$, we will assume throughout that the optimizer has access to a *stochastic gradient oracle* (SGO), i.e., a black-box mechanism that returns an estimate of the gradient of $f$ at the point of interest. Formally, when queried at a $x \in \mathcal{X}$, an SGO returns a *stochastic gradient* of the form

$$\mathsf{G}(x; \omega) = \nabla f(x) + \mathsf{Z}(x; \omega) \qquad \text{(SGO)}$$

where $\omega$ is a random seed drawn from some (complete) probability space $(\Omega, \mathcal{F}, \mathbb{P})$, and $\mathsf{Z}(x; \omega)$ denotes an umbrella error term intended to capture all sources of noise and randomness in the oracle. Our blanket assumption throughout will be that $\mathsf{G}$ is unbiased, that is,

$$\mathbb{E}[\mathsf{Z}(x; \omega)] = 0 \quad \text{for all } x \in \mathcal{X}. \qquad \text{(4)}$$

This is a bare-bones requirement which is standard for stochastic gradient methods. Other than that, in terms of magnitude, we will make the following assumption for $\mathsf{Z}$.

**Assumption 3.** The gradient noise term $\mathsf{Z} \colon \mathcal{X} \times \Omega \to \mathbb{R}^d$ of (SGO) has pointwise sub-Gaussian tails, i.e.,

$$\mathbb{E}[\exp(\langle \mathsf{Z}(x; \omega), p \rangle)] \leq e^{\frac{1}{2}\sigma(x)^2 \|p\|^2} \qquad \text{(SubG)}$$

for some *variance proxy* parameter $\sigma(x)$, $x \in \mathcal{X}$.

Assumption 3 simply means that the tails of $\mathsf{Z}$ are bounded as

$$\mathbb{P}(\|\mathsf{Z}(x; \omega)\| \geq t) \leq 2e^{-\frac{t^2}{2\sigma(x)^2}} . \qquad (5)$$

Importantly, the variance proxy parameter $\sigma(x)$ in (5) is state-dependent, so the variance of the noise may grow unbounded for large $x$. We will explore the precise ramifications of this assumption in the next section.

During implementation, $\mathsf{G}$ is typically queried along a sequence of points $x_t \in \mathcal{X}$, $t = 1, 2, \ldots$, with random seeds $\omega_t$ drawn i.i.d. from $\Omega$ according to $\mathbb{P}$. In this way, we get a sequence of stochastic gradients of the form

$$\hat{g}_t = \mathsf{G}(x_t; \omega_t) = \nabla f(x_t) + Z_t \qquad (6)$$

where $Z_t$ is a martingale difference sequence measuring the noise in the oracle at time $t$. For posterity, in order to formalize what is and what isn't random in (6), we will write $\mathcal{F}_t$ for the history of $x_t$ up to time $t$ (inclusive), so that $x_t$ is $\mathcal{F}_t$-measurable, while $\omega_t$, $\hat{g}_t$, and $Z_t$ are not.

**2.3. Discussion on the assumptions.** To illustrate the range of validity of Assumptions 1–3, consider the regularized empirical risk minimization (ERM) objective

$$f(x) = \frac{1}{N} \sum_{i=1}^{N} \ell(x; \xi_i) + \frac{\lambda}{2} \|x\|^2 \qquad \text{(ERM)}$$

where $\xi_i$, $i = 1, 2, \ldots, N$, is a set of training data points, $\ell(x; \xi)$ is the loss of model $x$ against datum $\xi$, and $\lambda > 0$ is a regularization parameter. In this finite-sum setting, the most common mechanism to estimate the gradient of $f$ is to sample a random minibatch $\omega \subseteq \{\xi_1, \ldots, \xi_N\}$ of fixed size $|\omega| = B \ll N$, and consider the minibatch gradient

$$\mathsf{G}(x; \omega) = \frac{1}{|\omega|} \sum_{\xi \in \omega} \nabla \ell(x; \xi) + \lambda x . \qquad (7)$$

Standard examples of this framework are neural networks with smooth activations and linear models with non-convex loss functions, cf. [22, 46, 52] and works cited therein. Accordingly, under standard choices for $\ell$ (squared / logistic loss, etc.), Assumptions 1 and 2 are trivially satisfied—the former thanks to the Tikhonov regularization term, the latter because $\ell$ grows at most polynomially at infinity. As for Assumption 3, taking $B = 1$ for simplicity (the general case is similar) and abusing notation to write $\omega \leftarrow \xi$, we get

$$\|\mathsf{Z}(x; \xi)\| = \left\| \frac{1}{N} \sum_{i=1}^{N} \nabla \ell(x; \xi_i) - \nabla \ell(x; \xi) \right\|$$

$$\leq \frac{1}{N} \sum_{i=1}^{N} \|\nabla \ell(x; \xi_i) - \nabla \ell(x; \xi)\| = \mathcal{O}(r(x)) \quad (8)$$

if $\|\nabla \ell(x; \xi)\| = \mathcal{O}(r(x))$ for all $\xi$ (so $r(x)$ can be chosen to grow at most polynomially in $x$). By a routine application of Höffding's lemma, this implies that $\log \mathbb{E}[\exp(\langle \mathsf{Z}(x; \omega), p \rangle)] = \mathcal{O}(\|p\|^2 r(x)^2)$, so Assumption 3 holds with $\sigma(x) = \mathcal{O}(r(x))$.

We detail the role of this scale function and the interplay between the growth of $f(x)$ and $\sigma(x)$ in the next section.

## 3 Taming SGD and first results

As we noted before, a crucial element in the convergence analysis of (SGD) is the $L$-smoothness assumption (LS): This condition ensures that gradients grow at most linearly at infinity, a property which is in turn essential for controlling the algorithm's iterates and establishing convergence. By contrast, under the more general assumption of $(L_0, L_1)$-smoothness, this growth condition breaks down, so the algorithm's updates must be controlled in a different manner—typically through some combination of gradient clipping and renormalization techniques.

These mechanisms allow us to recover the ergodic convergence guarantees of "randomly stopped" variants of (SGD) à la Ghadimi & Lan [25], producing a random output point $\bar{x}_T$ with $\mathbb{E}[\|\nabla f(\bar{x}_T)\|^2] = \mathcal{O}(\varepsilon)$ after $T = \mathcal{O}(1/\varepsilon^2)$ iterations on average [42, 45, 83] . However, finer properties of (SGD)—like the convergence of the generated trajectories or the avoidance of non-minimizing critical points—remain highly elusive in this setting.

We address these questions via an adaptive rescaling technique known as "taming". To build intuition for the method, we start with a basic taming scheme, which we analyze here under a slight refinement of Assumption 3; for a more sophisticated algorithm and analysis, see Section 4.

**3.1. Tamed SGD.** To make the motivation clearer and connect our approach with the literature, we first discuss the deterministic gradient case $\hat{g}_t = \nabla f(x_t)$, and we reinstate the stochastic regime after this first discussion.

To begin, recall that the optimal step-size choice for gradient descent in the class of $L$-smooth functions is of the form $\gamma = \alpha/L$ for some $\alpha \in (0, 2)$, leading to the update rule

$$x_{t+1} = x_t - (\alpha/L)\nabla f(x_t) . \qquad (9)$$

If we regard $(L_0, L_1)$-smoothness as "vanilla" Lipschitz smoothness with $L$ replaced by $L_0 + L_1 \|\nabla f(x)\|$, a plausible analogue of (9) for $(L_0, L_1)$-smooth functions would be

$$x_{t+1} = x_t - \frac{\alpha}{L_0 + L_1 \|\nabla f(x)\|} \nabla f(x) . \qquad (10)$$

This update is commonly referred to as $(L_0, L_1)$ *gradient descent*, and it is known to exhibit the same convergence properties as ordinary gradient descent for $L$-smooth functions, see e.g., Li et al. [45, Thms. 4.2 and 4.3] and Gorbunov et al. [29, Thm. 3.3] for the (strongly) convex case, and Li et al. [45, Thm. 5.2] for non-convex problems.

Since the constants $L_0$ and $L_1$ are not always known, a more flexible update rule would be the *normalized gradient descent* algorithm

$$x_{t+1} = x_t - \frac{\alpha_t}{\beta + \|\nabla f(x_t)\|} \nabla f(x_t) \qquad \text{(NGD)}$$

for tunable $\alpha_t, \beta > 0$.[2] In a similar vein, a variant of (NGD) which is very popular in practice is the *clipped gradient descent* algorithm

$$x_{t+1} = x_t - \frac{\alpha_t}{\max\{\beta, \|\nabla f(x_t)\|\}} \nabla f(x_t) \qquad \text{(CGD)}$$

which "clips" the renormalization factor of (NGD) when $\nabla f(x_t)$ drops below the threshold parameter $\beta$. This clipping technique was introduced by Pascanu et al. [62, 63] as an efficient way to counter exploding gradients in (SGD) and to improve the method's stability and performance in neural network training, cf. [1, 27] and references therein.

The starting point of our approach is the observation that all these algorithms can be seen as special instances of the more general *tamed gradient descent* scheme

$$x_{t+1} = x_t - \frac{\gamma_t}{1 + \lambda_t r(x_t)} \nabla f(x_t) \qquad \text{(TGD)}$$

where $\lambda_t > 0$ is a variable *taming parameter* and $r(x) \geq 0$ is a suitably chosen continuous coercive *scale function*. From an implementation standpoint, the most immediate choice for this scale function would be to ensure that $\|\nabla f(x)\| = \mathcal{O}(r(x))$ in order to dampen excessively large gradients and stabilize the iterates of the method. In so doing, we have:

(a) The $(L_0, L_1)$ update (10) is recovered by taking $\gamma_t \leftarrow \eta/L_0$, $\lambda_t \leftarrow L_1/L_0$, and $r(x) = \|\nabla f(x)\|$.

(b) (NGD) is obtained by setting $\gamma_t \leftarrow \alpha_t/\beta$, $\lambda_t \leftarrow 1/\beta$, and $r(x) = \|\nabla f(x)\|$.

(c) (CGD) is recovered by letting $\gamma_t \leftarrow \alpha_t/\beta$, $\lambda_t \leftarrow 1/\beta$, and $r(x) = \max\{\beta, \|\nabla f(x)\|\} - \beta$.

Thus, returning to the stochastic regime, if the optimizer can only access stochastic gradient (SG) information, we obtain the *tamed stochastic gradient descent* scheme

$$x_{t+1} = x_t - \frac{\gamma_t}{1 + \lambda_t r(x_t)} \hat{g}_t \qquad \text{(TSGD)}$$

---

[2]We follow here the terminology of Zhang et al. [84]; at the same time, we should note that (NGD) is also encountered with $\beta = 0$, cf. Chen et al. [17] and Vankov et al. [79].

with $\hat{g}_t$ generated by (SGO) as per (6). This method will be our main focus in this section, so some remarks are in order.

*Remark* 1. The terminology—and methodology—of "taming" has its roots in the numerical analysis of SDEs, where it has been used extensively to remedy the numerical instability of approximating solutions of SDEs with superlinearly growing gradients. More precisely, the solutions of the Langevin SDE

$$dX_t = -\nabla f(X_t) \, dt + \, dW_t \qquad \text{(LSDE)}$$

and the trajectories of its Euler–Maruyama discretization

$$x_{t+1} = x_t - \gamma \nabla f(x_t) + \sqrt{\gamma} Z_t \qquad \text{(EM)}$$

are known to diverge away from each other at an exponential rate when $\nabla f$ grows superlinearly [35] (in the above, $W_t$ is a standard Brownian motion in $\mathbb{R}^d$ and $Z_t$ is a Gaussian random vector with zero mean and unit covariance). In this context, "taming" consists of constructing a modified drift coefficient $b_\lambda(x)$ for (LSDE) such that $\|b_\lambda(x)\| = \mathcal{O}(\|x\|)$ and $b_\lambda(x) \to -\nabla f(x)$ as $\lambda \to 0$. The taming factor of (TSGD) is a typical modification of this sort—cf. [30, 36, 71, 72] for its uses in numerical analysis, and [38, 39, 47, 51, 53–56, 60**?** ] for its applications to sampling. ✦

*Remark* 2. A subtle aspect of the tamed scheme (TSGD) is that, as stated, the scale factor $r(x)$ is deterministic, so it does not allow taming by a random factor like $\|\hat{g}_t\|$ (or any other factor that is not $\mathcal{F}_t$-measurable). To recover random normalization (or clipping) schemes of this type, one could consider *randomized* scale functions of the form $r(x; \omega)$ instead of $r(x)$; however, this choice introduces an unavoidable estimation bias which, if left unchecked, could prevent convergence to crit $f$ (see the related discussion in Chen et al. [16], Koloskova et al. [42] and references therein). This bias can be mitigated under stronger assumptions for the noise—e.g., boundedness, as in [83, 84]—but the analysis and results are much more transparent with a deterministic scale factor, so we do not treat this more general model here. ✦

**3.2. Analysis and results.** We are now in a position to move forward with our analysis of (TSGD). As we mentioned earlier, a major challenge here is the control of the gradient noise term. To that end, our approach will hinge on the observation that Assumption 3 implies the exponential moment bound

$$\log \mathbb{E}[e^{\|Z(x;\omega)\|}] \leq C_1 \sigma(x) + C_2 \sigma(x)^2 \qquad (11)$$

for suitable constants $C_1, C_2 > 0$ and all $x \in \mathcal{X}$. This is a key technical result which requires a delicate handling of the tail bound (5), and which we detail in Appendix B.

In view of (11), to relate this bound with the growth of $f$ in a way that is compatible with $(L_0, L_1)$-smoothness, we will make the following ABC-type assumption:

**Assumption 4.** The variance proxy of $\mathsf{Z}(x; \omega)$ is bounded as

$$\sigma(x)^2 \leq A[f(x) - \inf f] + B\|\nabla f(x)\|^2 + C \quad (\text{ABC}_\sigma)$$

for some $A, B, C \geq 0$ and all $x \in \mathcal{X}$.

*Remark* 3. By our standing assumptions for $f$, we have $\inf f = f_* > -\infty$, cf. Eq. (1a) in Section 2; obviously, if $\inf f = -\infty$, the bound ($\text{ABC}_\sigma$) becomes meaningless. ✦

*Remark* 4. The terminology "ABC" is a pointer to Khaled & Richtárik [40], who considered the condition

$$\mathbb{E}[\|\mathsf{G}(x; \omega)\|^2] \leq A[f(x) - f_*] + B\|\nabla f(x)\|^2 + C \quad (\text{ABC})$$

for some $A, B, C \geq 0$ and all $x \in \mathcal{X}$. This condition is not equivalent to ours—though it is implied by it, cf. Lemma B.4 in Appendix B. Different versions of (ABC) go back to Blum [10], Gladyshev [26], and Bertsekas & Tsitsiklis [9] (with some terms missing, or a slightly different scaling, depending on the precise version under study). For an excellent survey and an in-depth treatment of related conditions in the literature, see Alacaoglu et al. [2]. ✦

Assumption 4 suggests that the stochastic gradients entering the algorithm can be controlled by the RHS of ($\text{ABC}_\sigma$). Thus, going term-by-term, a sensible setting for the scale function of (TSGD) would be to take

$$r(x) = \Omega\left(\max\left\{1, \sqrt{f(x) - f_*}, \|\nabla f(x)\|\right\}\right) \quad (12)$$

implying in turn—by unfolding ($\text{ABC}_\sigma$)—that $\sigma(x) = \mathcal{O}(r(x))$. Importantly, (12) concerns only the asymptotic growth of $f(x)$ and $\|\nabla f(x)\|$, so it is easy to meet in practice.

Below we provide some typical examples of optimization problems, where the objective function provides a precise characterization for the scale function $r(x)$.

1. **Feedforward Neural Network.** The optimization problem of training a deep neural network with square loss, has the following form

$$\min f(x) = \sum_{i=1}^{N} f_i(x)/N, \quad f_i(x) = (h(w_i; x) - y_i)^2 / 2$$
$$(13)$$

where $h(w; x)$ is the prediction model and $(w_i, y_i)_{i \leq N}$ the couples of input-output data. If the model has depth $L$ and is positively homogeneous (which is the case for most of the common Lipschitz activation functions, e.g. ReLU, GELU, softplus, sigmoid, Swish,...), then one can take a scale function of the form $r(x) = 1 + \|x\|^{2L-1}$.

2. **Phase retrieval.** The standard Phase retrieval inverse problem has the following form

$$\min f(x) = \sum_{i=1}^{N} f_i(x)/N, \quad f_i(x) = \left((w_i^\mathsf{T}x)^2 - y_i\right)^2 / 4,$$
$$(14)$$

where $(w_i)_{i \leq N}$ are the sampling coefficients and $(y_i)_{i \leq N}$ the observed measurements. In such problems, the objective function is a quartic polynomial in the optimization parameter, yielding a scale function $r(x) = 1 + \|x\|^3$.

3. **Matrix factorization/completion.** In this case the optimization problem can be usually expressed as

$$\min f(U, V) = \|UV^\mathsf{T} - M\|_F^2 / 2 \quad (15)$$

where the (coupled) optimization variable is $(U, V)$. In this case, the objective function is quartic in $(U, V)$, which naturally leads to a scale function of the form $r(U, V) = 1 + \|(U, V)\|_F^3$.

In the black-box case, where there is no structural information about $f$ whatsoever, the sole assumption of $(L_0, L_1)$-smoothness ensures that $f$ and $\nabla f(x)$ grow at most exponentially in $x$, so, as a last resort, one can take $r(x) = e^{\|x\|}$ (see Lemma B.1 and Remark 5 in Appendix B). We stress however that, in most cases of practical interest, this would be a needlessly pessimistic choice.

With all this in hand, we obtain the following almost sure convergence result for (TSGD).

**Theorem 1.** *Suppose that Assumptions 1–4 hold. Assume further that (TSGD) is run with a scale function $r(x)$ as per (12), and bounded step-size/taming parameters such that*

$$\sum_{t=1}^{\infty} \gamma_t = \infty, \quad \sum_{t=1}^{\infty} \frac{\gamma_t^2}{\lambda_t} < \infty \quad \text{and} \quad \lim_{t \to \infty} \frac{\gamma_t}{\lambda_t} = 0. \quad (16)$$

*Then, with probability 1, we have $\lim_{t \to \infty} \nabla f(x_t) = 0$. In particular, if $\gamma_t \propto 1/t^p$ and $\lambda_t \propto 1/t^q$, $p, q \in [0, 1]$, $x_t$ converges to $\text{crit } f$ (a.s.) as long as $(q + 1)/2 < p \leq 1$.*

To the best of our knowledge, Theorem 1 is the first almost sure convergence result in the literature for functions that are locally—but not *globally*—Lipschitz smooth. In the non-convex setting, existing results require either a bounded iterates assumption ($\sup_t \|x_t\| < \infty$ with probability 1) [5, 12, 43, 48, 50] or a global Lipschitz smoothness assumption [9, 59, 81] (and in all cases, stochastic gradients with uniformly bounded variance). In this regard, Theorem 1 broadens substantially the class of functions where convergence is guaranteed—encompassing in particular deep learning objectives that typically exhibit superlinear gradient growth. This flexibility is a central advantage of the taming approach.

We prove Theorem 1 in Appendix C and only provide a sketch of proof below. The first step is relatively straightforward and involves deriving a suitable (one-step) descent-like inequality under $(L_0, L_1)$-smoothness. However, in contrast to the $L$-smooth case, the derived descent inequality involves a term that scales exponentially with the gradient input to (TSGD). To tame this term, we employ a series

of sharp bounds for random variables with sub-Gaussian moments, which appear to be new in the literature. Then, coupling Assumption 4 with a carefully tailored schedule for $\gamma_t$ and $\lambda_t$—which is where the summability conidtions (16) originate—a second series of exponential moment bounds allows us to show that the evolution of $f(x_t)$ under (TSGD) forms an "almost supermartingale", that is, a supermartingale up to a summable error term. At this point, by a direct combination of the Robbins–Siegmund theorem [69] and the coercivity of $f$, it follows that the iterates $x_t$ of (TSGD) are bounded with probability 1. Theorem 1 then follows by leveraging the *local* Lipschitz continuity of $\nabla f$ along with a lemma on numerical sequences by Orabona [61], itself a simplification of an arduous string of estimations by Bertsekas & Tsitsiklis [9].

The noise assumption (ABC$_\sigma$) plays a crucial role in our analysis of (TSGD), as it couples the (central) moments of $\hat{g}_t$ with the growth of $f$ and $\|\nabla f\|$, which is what is ultimately required to establish the "almost submartingale" property for $f(x_t)$. Without Assumption 4, showing that the iterates of the method are bounded (a.s.) is considerably more difficult; we address this issue in the next section.

## 4 Dissipative taming

We now return to the general, umbrella setting of Assumption 3, and we present a more flexible taming scheme for (SGD) that does not require (ABC$_\sigma$) as an additional assumption. The starting point of our approach is the ansatz that, if $x_t$ converges to crit $f$, it is much more likely to encounter very large gradients in the earlier stages of the method rather than later ones. As such, to bootstrap this ansatz, we would require a mechanism that expressly dissipates large gradients that occur far from crit $f$, but whose influence decays with time (in order to avoid introducing a lasting bias in the algorithm's gradient steps).

A direct way to achieve this would be to introduce an explicit regularization term in (TSGD), as per the update rule

$$x_{t+1} = x_t - \frac{\gamma_t}{1 + \lambda_t r(x_t)} \hat{g}_t - \gamma_t a_t x_t \tag{17}$$

for some suitable regularization parameter $a_t > 0$. From a qualitative viewpoint, the term $a_t x_t$ has the desired coercivity effect as it provides an explicit drift towards the origin. From a quantitative standpoint however, the regularization schedule $a_t$ would have to be set "just right": too small and the iterates might escape, too large and the dissipative term might overcompensate and introduce an asymptotic bias relative to (SGD).

To overcome this roadblock, we consider instead an *implicit* regularization scheme where the amount of dissipation is directly coupled to the taming mechanism. This yields the

*dissipatively tamed stochastic gradient descent* algorithm

$$x_{t+1} = x_t - \gamma_t \frac{\hat{g}_t - a x_t}{[1 + \lambda_t r(x_t)^{1/\varepsilon}]^\varepsilon} - \gamma_t a x_t, \quad \text{(DTSGD)}$$

where the parameters $a$ and $\varepsilon$ can be tuned freely by the optimizer. In more detail, setting aside for the moment the role of $\varepsilon$, the iterates of (DTSGD) can be rewritten as

$$x_{t+1} = x_t - \frac{\gamma_t \hat{g}_t}{[1 + \lambda_t r(x_t)^{1/\varepsilon}]^\varepsilon} - \gamma_t a_t x_t \tag{18}$$

where

$$a_t = \frac{[1 + \lambda_t r(x_t)^{1/\varepsilon}]^\varepsilon - 1}{[1 + \lambda_t r(x_t)^{1/\varepsilon}]^\varepsilon} a. \tag{19}$$

In this regard, the main difference between (DTSGD) and (17) is that the regularization weight in the latter does not have to be guessed at the outset: instead, it grows or shrinks adaptively based on the amount of taming in the process. When $x_t$ grows large, the regularization weight parameter $a_t$ is approximately equal to $a$, so the scheme is *strongly dissipative*; by contrast, when $x_t$ is small for large $t$, $a_t$ is near-zero, so the scheme is *weakly dissipative*.

With all this in mind, and motivated by the discussion of the empirical risk minimization problem in Section 2.3, we will assume in the rest of this section that (DTSGD) is run with a scale function $r(x)$ such that

$$r(x) = \Omega(\max\{\|\nabla f(x)\|, \sigma(x)\}). \tag{20}$$

As we discussed in Section 3, in the context of (ERM), $\|\nabla f(x)\|$ and $\sigma(x)$ exhibit the same growth, so $r(x)$ can be readily estimated from the structure of the problem's loss function $\ell$. For example, in the case of an $L$-layer neural network with hinge losses and Lipschitz activation functions (ReLU, softplus, Swish, sigmoid, ...), one can simply take $r(x) = \|x\|^L$. More generally, by the $(L_0, L_1)$ smoothness condition (L$_0$L$_1$), gradients grow at most exponentially, so we impose throughout the pessimistic bound

$$r(x) = \exp(\mathcal{O}(\|x\|)). \tag{21}$$

To streamline our presentation, we will treat (21) as an operational ceiling, but we stress that it would be overkill to select $r(x) = \exp(\mathcal{O}(\|x\|))$ in practice: as in the case of (ERM), the vast majority of problems of interest in data science and machine learning have sufficient structure to allow for a much more principled choice for $r$.

As for the exponent $\varepsilon$, this is an a posteriori scaling trick which allows us greater flexibility in setting the step-size and taming parameters of the method. With all this in mind, we have the following almost sure convergence result.

**Theorem 2.** *Suppose that Assumptions 1–3 hold. Assume further that* (DTSGD) *is run with a scale function $r(x)$ as*

*above, and step-size / taming parameters such that*

$$\sum_{t=1}^{\infty} \gamma_t = \infty, \quad \sum_{t=1}^{\infty} \gamma_t \lambda_t < \infty \quad and \quad \sum_{t=1}^{\infty} \frac{\gamma_t^2}{\lambda_t^{2\varepsilon}} < \infty. \quad (22)$$

*Then, with probability $1$, we have $\lim_{t \to \infty} \nabla f(x_t) = 0$. In particular, if $\gamma_t \propto 1/t^p$ and $\lambda_t \propto 1/t^q$, then $x_t$ converges to crit $f$ (a.s.) as long as $0 \le 1 - p < q < (p - 1/2)/\varepsilon$.*

The proof of Theorem 2 is deferred to Appendix D. While the main lines of the proof follows those of Theorem 1, the absence of Assumption 4 (use of the extended descent lemma, i.e., Lemma B.1) on the stochastic oracle makes the associated analysis considerably more challenging in terms of the noise control. The extra regularization term $a_t$ in (19) which is used in the scheme (DTSGD), has exactly the role of regularizing even further the noise effect and allows to obtain useful *à priori* estimates on the (square) exponential moments of the iterates (cf. Lemma D.6), that are necessary to conclude with an "almost supermartingale" relation *à la* Robbins-Siegmund [69]. The last part of the proof follows the arguments presented in Orabona [61, Lemma 1] with some additional handling due to the regularization term $a_t$.

## 5 Avoidance analysis

In this last section, we turn to the sequend question we asked in the introduction, seek to see if it is possible to refine the convergence guarantee of Theorems 1 and 2 so as to exclude convergence to non-minimizing saddle points of $f$.

To make this statement precise, we say that $q \in$ crit $f$ is a *strict saddle point* of $f$ if $\lambda_{\min}(\nabla^2 f(q)) < 0$. More generally, following [59], we say that a smooth connected component $\mathcal{Q}$ of crit $f$ is a *strict saddle manifold* of $f$ if

(a) $\mathcal{Q}$ consists of strict saddle points, i.e., $\lambda_{\min}(\nabla^2 f(q)) < 0$ for all $q \in \mathcal{Q}$.

(b) There exist constants $c_{\pm} > 0$ such that, for all $q \in \mathcal{Q}$, the negative eigenvalues of $\nabla^2 f(q)$ are bounded above by $-c_- < 0$, and any positive eigenvalues thereof are bounded below by $c_+$ (if any such eigenvalues exist).

Condition (b) above is a technical requirement that ensures that the eigenspaces of $\nabla^2 f$ decompose smoothly along its negative, zero, and positive eigenvalues (again, if the latter exist). In this way, the $\mathcal{Q}$ admits three (smooth) vector bundle structures, one along the negative, unstable directions of $\nabla^2 f$, one along the integral manifold of the nullspace of $\nabla^2 f$ over crit $f$, and one along the stable directions of $\nabla^2 f$ (which could be empty).

The second condition that we will require is a refinement of our noise assumptions to avoid degeneracies and maintain control on the noise near crit $f$.

**Assumption 5.** The gradient noise term $\mathsf{Z} \colon \mathcal{X} \times \Omega \to \mathbb{R}^d$ of (SGO) satisfies the following conditions:

(a) *Uniform excitation:* There exists some $c > 0$ such that

$$\mathbb{E}[\langle \mathsf{Z}(x; \omega), u \rangle_+] \ge c \quad (23)$$

for all $x \in \mathcal{X}$ and all unit vectors $u \in \mathbb{R}^d$, $\|u\| \le 1$.

(b) *Boundedness near critical points:*

$$\sup_{x \in \mathcal{U}_c} \|\mathsf{Z}(x; \omega)\| < \infty \quad (24)$$

for some compact neighborhood $\mathcal{U}_c$ of crit $f$.

At a high level, Assumption 5(a) simply means that there is "enough noise" in the process to excite the unstable directions of $\nabla f^2(x)$; this could be relaxed to a "transversal excitation" requirement, where the noise is only required to excite those directions that are transverse to the non-positive eigenspacs of $\nabla f^2(x)$, but this is not important for our purposes and lies beyond the scope of our work. As for Assumption 5(b), this is simply a technical bound on the steps of the process near a saddle manifold: since crit $f$ is compact (by Assumption 1), it is easy to check that it is satisfied by (ERM).

Under this assumption, we obtain the following result:

**Theorem 3.** *Suppose that Assumptions 1, 3 and 5 hold, and let $x_t$ be the sequence of iterates generated by (TSGD) or (DTSGD), with step-size and taming parameters of the form $\gamma_t \propto 1/t^p$ and $\lambda_t \propto 1/t^q$, with $p \in (1/2, 1]$, $q \in (0, 1]$, and $p/2 < q \le 1$. If $\mathcal{Q}$ is a strict saddle manifold of $f$, we have $\mathbb{P}(x_t \to \mathcal{Q} \text{ as } t \to \infty) = 0$.*

Thus, applying Theorem 3 in tandem with Theorems 1 and 2 for taming and step-size schedules of the form $\gamma_t \propto 1/t^p$ and $\lambda_t \propto 1/t^q$ with $p \in (1/2, 1]$, $q \in (0, 1]$, we have:

1. With probability $1$, (TSGD) converges to crit $f$ and avoids strict saddle manifolds as long as

$$2/3 < p \le 1 \quad and \quad p/2 < q < 2p - 1. \quad (25a)$$

2. With probability $1$, (DTSGD) converges to crit $f$ and avoids strict saddle manifolds as long as

$$1 - p < q < (p - 1/2)/\varepsilon \quad and \quad p/2 < q \le 1 \quad (25b)$$

for some $\varepsilon \in (0, 1)$.

We plot the relevant parameter regions for different values of $\varepsilon$ in Fig. 3 in Appendix E. Importantly, for small values of $\varepsilon$, we see that (DTSGD) allows step-size schedules that are arbitrarily close to the standard Robbins–Monro boundary $p > 1/2$, whereas (TSGD) is constrained to $p > 2/3$.

The proof of Theorem 3 hinges on the observation that the iterative structure of both (TSGD) and (DTSGD) can be recast in the generalized Robbins–Monro template

$$x_{t+1} = x_t - \gamma_t(\nabla f(x_t) + Z_t + b_t) \quad (\text{RM})$$

where, under the ansatz that $x_t$ lies in the vicinity of crit $f$, the bias term $b_t$ of (RM) can be shown to be $\mathcal{O}(\lambda_t)$. With this basic estimate in place, the rest of our proof hinges on two basic pillars: The first is a deep differential-geometric construction, originally due to Benaïm & Hirsch [6] and Benaïm [5], which leverages the center stable manifold theorem to build a suitable Lyapunov function which "stratifies" the center/stable and unstable manifolds of a strict saddle manifold $\mathcal{Q}$ under the gradient flow of $f$. The second is a probabilistic estimate for conditionally positive stochastic processes due to Pemantle [64] and extended by Benaïm [5] which allows us to show that the probability of the noise acting against the gradient flow of $f$ and driving the process against the unstable directions of $\mathcal{Q}$ is zero. We provide all the necessary details in Appendix E.

## 6 Concluding remarks

Perhaps the most important take-away of our work is that, beyond its origins as a numerical integrator of unstable SDEs, taming provides a flexible and principled paradigm for stabilizing SGD in non-convex landscapes with steep, superquadratic growth. We find the method's almost sure convergence and saddle-point avoidance under $(L_0, L_1)$-smoothness—both results extending the scope of what can be shown in the context of generalized smoothness conditions—particularly appealing for the applicability of the method to deep learning loss landscapes, where saddle points are ubiquitous. In this regard, a natural—albeit highly challenging—research direction that arises is to examine whether taming can be combined with momentum-based optimizers—such as Adam—and whether it is possible to obtain trajectory-wide convergence rates (as opposed to ergodic guarantees). We leave these questions to the future.

## 7 Numerical experiments

Our goal in this section is to illustrate the behavior of (TSGD) on two simple synthetic experiments: *a)* a phase retrieval problem; and *b)* the training of a (regularized) deep diagonal neural network model. in order to complement our theoretical results. For the details of our setup and additional experiments, see Appendix F.

**7.1. Phase retrieval.** We consider the Phase retrieval inverse problem (14) with synthetically generated (Gaussian) random vectors $(w_i)_{i \leq n} \in \mathbb{R}^d$ and $x_* \in \mathbb{R}^d$ and the observations are set as $y_i = (w_i^\mathsf{T} x_*)^2$, with $d = 10$ and $N = 200$.

In Fig. 1, we illustrate the performance of (TSGD) in terms of success rate for the convergence criterion $\|\nabla f(x_t)\| \leq 0.001$, over 50 runs, for different maximum caps of number of iterations. We compare three instances of (TSGD) with $r(x) = \|x\|^3$, $r(x) = \|x\|^5$ and $r(x) = \exp(\|x\|)$ (the last two correspond to a more pessimistic choice for the scale

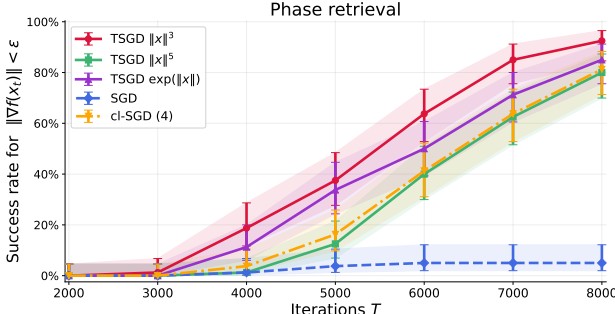

**Figure 1.** Success rate for achieving an accuracy of 0.001 (in terms of gradient) across different max number of iterations, over 80 runs. Here we compare three instances of (TSGD) with $r(x) = \|x\|^3$ (red), $r(x) = \|x\|^5$ (green) and $r(x) = \exp(\|x\|)$ (purple), with vanilla (SGD) (blue) and clipped stochastic (CGD) (orange).

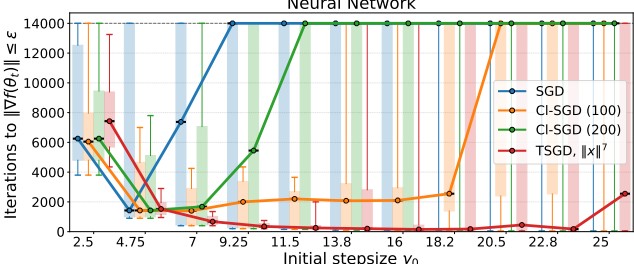

**Figure 2.** Number of iterations needed in order to achieve an accuracy of at least 0.005 for different initializations of the stepsize $\gamma_0$ when solving the regularized ERM problem with diagonal network (F.3). Here we compare (TSGD) (red) with clipped stochastic CGD with two clipping levels (orange & green) and vanilla (SGD) (blue). Each box (and whisker resp.) represents the interquartile (and min-max resp.) range over 50 independent runs and the thick colored lines connect the medians of each box.

function $r(x)$) together with vanilla (SGD) and stochastic (CGD). Overall, we observe that the "tight" version of (TSGD) (with $r(x) = \|x\|^3$) performs better than all the other methods both at low and high maximum number of iterations.

**7.2. Diagonal Neural Network.** We consider the problem of training a (regularized) diagonal Neural Network as in (13), with $f_i(x) = \frac{1}{2} (h(w_i; x) - y_i)^2 + \frac{\mu}{2} \|x\|_F^2$ where $\mu > 0$ and the prediction model is given by $h(w; x) = \sum_{k=1}^d \left( \prod_{\ell=1}^L x_{\ell k} \right) w_k$, with $d = 5$, $N = 128$ and $L = 4$. In this experiment, we illustrate the behavior of (TSGD), in terms of stability to the choice of the initialization $\gamma_0$ of the step-size (see e.g. [4, 77]), and compare it with vanilla (SGD) and stochastic (CGD) with two different clipping levels. The results are reported in Figure 2 (see also Fig. 5 in Appendix F). From Fig. 2, we can observe that for large values of $\gamma_0$, the (TSGD) is more stable, maintaining a good performance, even when the rest of the other methods, fail to achieve the convergence criterion.

## Acknowledgments

This research was supported in part by the French National Research Agency (ANR) in the framework of the PEPR IA FOUNDRY project (ANR-23-PEIA-0003). VA, IL and PM are also members of the Archimedes Research Unit/Athena RC, and were partially supported by project MIS 5154714 of the National Recovery and Resilience Plan Greece 2.0 funded by the European Union under the NextGenerationEU Program.

## Impact Statement

This paper presents work whose goal is to advance the field of Machine Learning. There are many potential societal consequences of our work, none of which we feel must be specifically highlighted here.

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

# A  Further related work

In this Section we further discuss the related literature.

The theoretical study of gradient based methods under the $(L_0, L_1)$ smoothness has its origins in the seminal work of Zhang et al. [84] and provides a suitable relaxation to the standard smoothness (LS),allowing to deal with a considerably large class of function that are met in practice. In the subsequent works of [83] and [17, 79] (see also [29, 42]), several first-order relaxations of the original $(L_0, L_1)$ smoothness, such as (3)and (2), have been made, lifting the $C^2$ character of ($L_0L_1$). Lastly, further relaxations of ($L_0L_1$), namely generalized $\ell$-smoothness, have also been made in [45] (see also [78]) to encompass more general growth rate for the curvature (norm of the Hessian) with respect to the magnitude of the gradient, via a (not necessarily affine) function $\ell(\cdot)$.

The technique of clipping on the vanilla (stochastic) gradient descent has been introduced in [7, 31, 62, 63] (see also [19, 28, 57, 84] and references therein) as an efficient way to prevent gradient explosion, when training deep neural networks and therefore stabilize its behavior. In particular, clipped and normalized variants of (stochastic) gradient methods have been proven to be some of the most effective schemes to deal with $(L_0, L_1)$ smooth objectives. In the deterministic non-convex setting the authors in [83, 84] (see also [17, 42, 79]) provide iteration complexity bounds for $\|\nabla f(x_t)\|$ and $t^{-1} \sum_{s \geq 0} \|\nabla f(x_s)\|$, of order $\mathcal{O}(t^{-\frac{1}{2}})$ and $\mathcal{O}(t^{-1})$ (resp.) which are the same with the ones of gradient descent for smooth functions. Further improvements for clipped gradient descent for (strongly) convex objectives under the $(L_0, L_1)$ smoothness condition, in terms of complexity for $f(x_t) - \inf f$, are reported in Koloskova et al. [42, Theorems 2.3 & 2.5] and Gorbunov et al. [29, Theorem 3.3] and can be further coupled with acceleration techniques and Polyak step-size policies Takezawa et al. [76, Theorem 4], Gorbunov et al. [29, Theorems 5.2 & 4.1], Vankov et al. [79, Theorems 5.1 & 6.1] and Li et al. [45, Theorem 4.4]. These results were further extended for general $\ell$-smooth functions both in convex Li et al. [45, Theorems 4.2, 4.3 & 4.4] and non-convex setting Li et al. [45, Theorem 5.2] and Tyurin [78, Theorem 5.1].

In the stochastic setting the first convergence guarantees for stochastic versions of clipped (SGD), under $(L_0, L_1)$ smoothness were established in [84] and [83] (see also [82]) both for the time average of $\mathbb{E}[\|\nabla f(x_t)\|]$, as also for $\|\nabla f(x_t)\|$ with high probability. A pivotal assumption for these results was the uniformly bounded noise almost surely, which however may not hold in practice. In this regard, in Koloskova et al. [42, Theorems 3.1 & 3.2] the authors i) show that beyond uniform noise boundedness conditions the clipped SGD inherits unavoidable bias towards (a set of) of points that are not necessarily critical (see also the related discussion on [73]) and ii) quantify the rate of convergence for $\mathbb{E}[\|\nabla f(x_t)\|]$ over the best iterate, under a bounded variance assumption. Similar results for normalized (SGD) (with momentum) for the time average of $\mathbb{E}[\|\nabla f(x_t)\|]$ can be found in Zhao et al. [85, Theorem 2] and Hübler et al. [34, Theorem 14]. Convergence guarantees in high probability, via stopping time arguments, were also deduced for (SGD) under bounded variance of the noise Li et al. [45, Theorem 5.3] and signSGD with momentum [18, Theorem 1], as also in Faw et al. [23, Theorem 26] for AdaGrad-Norm [75] under a relaxed affine variance condition on the noise. Further convergence results in high probability for clipped SGD under light-tail (heavy-tail resp.) noise assumptions in a convex setting can be found in [?] ([?] resp.). Finally, clipped SGD was generalized in [66] to non-euclidean metrics by combining steepest descent and conditional gradient approaches under the assumption of $(L_0 - L_1)$ smoothness and bounded variance, where the authors achieved an order optimal rate in expectation [66, Theorem 4.9].

In a similar spirit to our work there have been few other efforts to use the taming technology in optimization, mainly through sampling. Based on the notion that for an objective function $f$ the invariant measure of a Langevin SDE with drift $-\nabla f$ concentrates around the minimum values of $f$ in the small temperature regime ([67]) there has been some effort to use Langevin-based samplers such as variates of GLD and SGLD alogrithms as optimizers. More specifically the first result for variants of SGLD beyond gradient Lipschitz continuity was achieved in [51] where the author provide convergence rates in expectation, and improved experimental results have been achieved in [47] and [?] while for GLD variants the first convergence guarantees were given in [55] and later improved in [56]. It should be noted that all these assume that gradient of the objective function grows at most polynomially, contrary to our work where exponential growth is allowed. Finally, another work where a tamed algorithm was used purely as an optimizer is in [21] where the authors propose a tamed version of SGD algorithm for Hilbert spaces and obtain results in expectation, however their the theoretical framework assumes $L-$smoothness and strong convexity.

# B  Auxiliary lemmas and results

In this section we provide some general technical tools and lemmas that are used for the proofs of the main results.

**B.1. General estimates.** The next Lemma provides a pivotal relation for the main analysis under Assumption 2.

**Lemma B.1.** *[17, Prop. 1]. Let $f : \mathbb{R}^d \to \mathbb{R}$ be a function satisfying Assumption 2. Then it holds*

$$f(x') \le f(x) + \langle \nabla f(x), x' - x \rangle + \frac{L_0 + L_1 \|\nabla f(x)\|}{2} e^{L_1 \|x' - x\|} \cdot \|x' - x\|^2 . \tag{B.1}$$

*for all $x, x' \in \mathbb{R}^d$.*

*In particular, for all $x_0 \in \mathbb{R}^d$, it holds*

$$\|\nabla f(x)\| \le \begin{cases} \left( \|\nabla f(x_0)\| + \dfrac{L_0}{L_1} \right) e^{L_1 \|x - x_0\|} - \dfrac{L_0}{L_1} & \text{if } L_1 > 0 \\ \|\nabla f(x_0)\| + L_0 \|x - x_0\| & \text{if } L_1 = 0 \end{cases} \tag{B.2}$$

*and*

$$f(x) \le \begin{cases} |f(x_0)| + \dfrac{L_1 \|\nabla f(x_0)\| + L_0}{L_1^2} e^{L_1 \|x - x_0\|} - \dfrac{L_0}{L_1} \|x - x_0\| - \dfrac{L_1 \|\nabla f(x_0)\| + L_0}{L_1^2} & \text{if } L_1 > 0 \\ |f(x_0)| + \|\nabla f(x_0)\| \|x - x_0\| + \dfrac{L_0}{2} \|x - x_0\|^2 & \text{if } L_1 = 0 \end{cases} \tag{B.3}$$

*for all $x \in \mathbb{R}^d$.*

*Remark* 5. Under Assumption 2, the estimates (B.2) and (B.3) in Lemma B.1, provide a direct worst-case upper bound $r(x)$ on the growth of $\|\nabla f(x)\|$ and $f(x)$ as $\|x\| \to +\infty$, i.e. $\|\nabla f(x)\| = \mathcal{O}(r(x))$ and $f(x) = \mathcal{O}(r(x))$ which is of the form $r(x) = 1 + e^{L_1 \|x\|}$.

*Proof of Lemma B.1.* Relation (B.1), has been already proved in [17, Prop. 1]. Next we provide the proof for relations (B.2) and (B.3).

Let $x_0 \in \mathbb{R}^d$ be a reference point such that $C_0 = \|\nabla f(x_0)\| < +\infty$ and for all $x \ne x_0$, define the curve $\gamma : [0, \|x - x_0\|] \to \mathbb{R}^d$, such that $\gamma(t) = x_0 + t \frac{x - x_0}{\|x - x_0\|}$

From the Fundamental Theorem of Calculus, we have:

$$\nabla f(\gamma(t)) = \nabla f(x_0) + \int_0^t \left\langle \nabla^2 f(\gamma(s)), \frac{x - x_0}{\|x - x_0\|} \right\rangle ds \tag{B.4}$$

By taking norms and using the Cauchy-Schwarz inequality, we find

$$\begin{aligned} \|\nabla f(\gamma(t))\| &\le \|\nabla f(x_0)\| + \int_0^t \|\nabla^2 f(\gamma(s))\| ds \\ &\le \|\nabla f(x_0)\| + \int_0^t \left( L_0 + L_1 \|\nabla f(\gamma(s))\| \right) ds \end{aligned} \tag{B.5}$$

where in the second line, we used Assumption 2 ($(L_0, L_1)$-smoothness).

By using Grönwall's inequality in (B.5) for the function $\|\nabla f(\gamma(t))\| + \frac{L_0}{L_1}$, we find [3]

$$\|\nabla f(\gamma(t))\| + \frac{L_0}{L_1} \le \left( \|\nabla f(x_0)\| + \frac{L_0}{L_1} \right) e^{L_1 t} \tag{B.6}$$

By choosing $t = \|x - x_0\|$ in (B.6), it follows

$$\|\nabla f(x)\| \le \left( \|\nabla f(x_0)\| + \frac{L_0}{L_1} \right) e^{L_1 \|x - x_0\|} - \frac{L_0}{L_1} \tag{B.7}$$

---

[3] Here we assume that $L_1 > 0$. The case $L_1 = 0$, reduces to the standard Lipschitz gradient assumption and gives $\|\nabla f(\gamma(t))\| = \|\nabla f(x_0)\| + L_0 t$ and $f(\gamma(t)) \le |f(x_0)| + \|\nabla f(x_0)\| t + \frac{L_0}{2} t^2$.

In the same way, by using the Fundamental Theorem of Calculus for $f$ and the Cauchy-Schwarz inequality, it follows [3]

$$f(x) \leq |f(x_0)| + \int_0^{\|x-x_0\|} \|\nabla f(\gamma(t))\| dt$$

$$|f(x_0)| + \frac{\|\nabla f(x_0)\| + \frac{L_0}{L_1}}{L_1} \left(e^{L_1\|x-x_0\|} - 1\right) - \frac{L_0}{L_1}\|x - x_0\|$$

(B.8)

■

The following technical Lemma can be found in [61] (see also [58, Lemma $A$.5]) and basically consists of a simplification of the analysis made in [9].

**Lemma B.2.** *[61, Lemma 1]. Let $\{a_t\}_{t\geq 1}$ and $\{\eta_t\}_{t\geq 1}$ be two non-negative sequences and $\{v_t\}_{t\geq 0}$ a sequence of vectors in $\mathbb{R}^d$, such that $\sum_{t=1}^{\infty} \eta_t a_t^2 < \infty$ and $\sum_{t=1}^{\infty} \eta_t = \infty$, and $\|\sum_{t=1}^{\infty} \eta_t v_t\| < \infty$. Let also $L \geq 0$ such that*

$$|a_{t+h} - a_t| \leq L \left(\sum_{s=t}^{t+h} \eta_s a_s + \|\sum_{s=t}^{t+h} \eta_s v_s\|\right)$$

(B.9)

*Then, $a_t$ converges to $0$.*

## B.2. Subgaussian random vectors.

The next two lemmas provide some useful results on subgaussian random vectors.

**Lemma B.3.** *Let $X \in \mathbb{R}^d$ be a subgaussian random vector with parameter $\sigma$. Then for every $\eta > 0$ there exists an absolute constant $C > 0$ such that*

$$\mathbb{E}[\exp(\eta\|X\|)] \leq \exp\left(C\eta\sigma\sqrt{d} + C\eta^2\sigma^2\right).$$

*Proof.* By the definition of a subgaussian vector and standard concentration results (see e.g. [80, Theorem 3.1.1]), there exist absolute constants $c, C > 0$ such that for all $t \geq 0$,

$$\mathbb{P}\left(\|X\| \geq C\sigma(\sqrt{d} + t)\right) \leq 2\exp(-ct^2).$$

(B.10)

Let $a := C\sigma\sqrt{d}$. Using the identity

$$\mathbb{E}[e^{\eta Y}] = 1 + \eta \int_0^{\infty} e^{\eta t} \mathbb{P}(Y \geq t) \, dt, \qquad Y \geq 0,$$

we obtain

$$\mathbb{E}[e^{\eta\|X\|}] \leq e^{\eta a} + \eta \int_a^{\infty} e^{\eta t} \mathbb{P}(\|X\| \geq t) \, dt.$$

Applying (B.10), for $t \geq a$,

$$\mathbb{P}(\|X\| \geq t) \leq 2\exp\left(-c\frac{(t-a)^2}{\sigma^2}\right).$$

Therefore,

$$\mathbb{E}[e^{\eta\|X\|}] \leq e^{\eta a} + 2\eta \int_a^{\infty} \exp\left(\eta t - c\frac{(t-a)^2}{\sigma^2}\right) dt.$$

Making the change of variables $s = t - a$ yields

$$\mathbb{E}[e^{\eta\|X\|}] \leq e^{\eta a} + 2\eta e^{\eta a} \int_0^{\infty} \exp\left(\eta s - c\frac{s^2}{\sigma^2}\right) ds.$$

Completing the square, we obtain

$$\eta s - c\frac{s^2}{\sigma^2} = -\frac{c}{\sigma^2}\left(s - \frac{\eta\sigma^2}{2c}\right)^2 + \frac{\eta^2\sigma^2}{4c},$$

and hence

$$\int_0^\infty \exp\left(\eta s - c\frac{s^2}{\sigma^2}\right) ds \leq \exp\left(\frac{\eta^2\sigma^2}{4c}\right) \int_{-\infty}^\infty \exp\left(-\frac{c}{\sigma^2}u^2\right) du \leq C\sigma \exp(C\eta^2\sigma^2).$$

Combining the estimates, we obtain

$$\mathbb{E}[e^{\eta\|X\|}] \leq \exp\left(C\eta\sigma\sqrt{d} + C\eta^2\sigma^2\right),$$

which proves the claim. ∎

**Lemma B.4.** *Assume that $X$ is a subgaussian random vector with parameter $\sigma$. Then, there exists some constant $C_p$, such that:*

$$\mathbb{E}[\|X\|^{2p}] \leq C_p\sigma^{2p}$$

*Proof.* By using [80, Proposition 6.2.1] there exists an absolute constant $c > 0$ such that

$$\mathbb{E}[\exp\frac{c\|X\|^2}{\sigma^2}] \leq C.$$

Setting $Y = \sqrt{c}\|X\|\sigma^{-1}$ on obtains that

$$\mathbb{E}[\|Y\|^{2p} = 2p\int_0^\infty t^{2p-1}P(\|Y\| \geq t)dt] \leq 2p\,\mathbb{E}[\int_0^\infty t^{2p-1}P(e^{\|Y\|^2}) \geq e^{t^2})dt] \leq \mathbb{E}[e^{\|Y\|^2}\int_0^\infty t^{2p-1}e^{-t^2}dt] \leq C'$$

which leads to

$$\mathbb{E}[\|X\|^{2p}] \leq C_p\sigma^{2p}$$

for some suitable constant $C_p$. ∎

# C  Convergence analysis of (TSGD)

In this section we provide the proof of Theorem 1 in Section 3. We recall that Algorithm (TSGD) can be expressed as follows

$$
\begin{aligned}
x_{t+1} &= x_t - \frac{\gamma_t}{1 + \lambda_t r(x_t)}\hat{g}_t \\
&= x_t - \gamma_t(g_t + \tilde{z}_t)
\end{aligned}
\tag{C.1}
$$

where

$$g_t := \frac{\nabla f(x_t)}{1 + \lambda_t r(x_t)} \quad \text{and} \quad \tilde{z}_t = \frac{Z_t}{1 + \lambda_t r(x_t)} \tag{C.2}$$

The following Lemma provides an estimate on the exponential moment of the noise term $\tilde{z}_t$.

**Lemma C.1.** *Suppose that Assumptions 1–4 hold true and let $\{x_t\}_{t\geq 0}$ be the sequence generated by Algorithm TSGD with $r(x)$ satisfying (12). Then, there exists some constant $C \geq 0$, such that*

$$\mathbb{E}_t\left[e^{2L_1\gamma_t\|\tilde{z}_t\|}\right] \leq e^{2c\frac{\gamma_t}{\lambda_t}} \tag{C.3}$$

*Proof.* By using the definition of $\tilde{z}_t = \frac{Z_t}{1+\lambda_t r(x_t)}$ and applying Lemma B.3 with $\eta = \frac{2L_1\gamma_t}{1+\lambda_t r(x_t)}$ it holds:

$$\mathbb{E}_t\left[e^{2L_1\gamma_t\|\tilde{z}_t\|}\right] = \mathbb{E}_t\left[e^{\frac{2L_1\gamma_t}{1+\lambda_t r(x_t)}\|Z_t\|}\right] \leq \exp\left(c_1\frac{2L_1\gamma_t\sigma(x_t)}{1+\lambda_t r(x_t)} + c_2\frac{4L_1^2\gamma_t^2\sigma(x_t)^2}{(1+\lambda_t r(x_t))^2}\right) \tag{C.4}$$

for some suitable constants $c_1$ and $c_2$. By using (ABC$_\sigma$) of Assumption 4 and (12), it holds $\sigma(x) \leq A\sqrt{f(x_t) - f_*} + B\|\nabla f(x_t)\| + C \leq r(x_t)$ and therefore from (C.4), it follows

$$\mathbb{E}_t\left[e^{2L_1\gamma_t\|\tilde{z}_t\|}\right] \leq \exp\left(C_1\frac{\gamma_t}{\lambda_t} + C_2\frac{\gamma_t^2}{\lambda_t^2}\right) \leq e^{2c\frac{\gamma_t}{\lambda_t}} \tag{C.5}$$

for some suitable constants $C_1$, $C_2$ and $c$. ∎

We are now in position to provide the full proof of Theorem 1, which we restate here for convenience.

**Theorem 1.** *Suppose that Assumptions 1–4 hold. Assume further that (TSGD) is run with a scale function $r(x)$ as per (12), and bounded step-size / taming parameters such that*

$$\sum_{t=1}^{\infty} \gamma_t = \infty, \quad \sum_{t=1}^{\infty} \frac{\gamma_t^2}{\lambda_t} < \infty \quad \text{and} \quad \lim_{t \to \infty} \frac{\gamma_t}{\lambda_t} = 0. \tag{16}$$

*Then, with probability $1$, we have $\lim_{t \to \infty} \nabla f(x_t) = 0$. In particular, if $\gamma_t \propto 1/t^p$ and $\lambda_t \propto 1/t^q$, $p, q \in [0, 1]$, $x_t$ converges to crit $f$ (a.s.) as long as $(q + 1)/2 < p \le 1$.*

***Proof of Theorem 1.*** By applying (B.1) in Lemma B.1 with $x' = x_{t+1}$ and $x = x_t$ and using (TSGD), we find:

$$\begin{aligned}
f(x_{t+1}) - f(x_t) &\le \langle \nabla f(x_t, x_{t+1} - x_t \rangle + \frac{(L_0 + L_1 \|\nabla f(x_t)\|)}{2} e^{L_1 \|x_{t+1} - x_t\|} \|x_{t+1} - x_t\|^2 \\
&= -\gamma_t \langle \nabla f(x_t), g_t \rangle - \gamma_t \langle \nabla f(x_t), \tilde{z}_t \rangle + \frac{(L_0 + L_1 \|\nabla f(x_t)\|)}{2} e^{L_1 \gamma_t \|g_t + \tilde{z}_t\|} \gamma_t^2 \|g_t + \tilde{z}_t\|^2 \\
&\le -\gamma_t \langle \nabla f(x_t), g_t \rangle - \gamma_t \langle \nabla f(x_t), \tilde{z}_t \rangle + (L_0 + L_1 \|\nabla f(x_t)\|) e^{L_1 \gamma_t (\|g_t\| + \|\tilde{z}_t\|)} \gamma_t^2 \left( \|g_t\|^2 + \|\tilde{z}_t\|^2 \right)
\end{aligned} \tag{C.6}$$

where in the last inequality, we used the triangle inequality (inside the exponential) and the convexity of $\|\cdot\|^2$ (i.e. $\|g_t + \tilde{z}_t\|^2 \le 2\|g_t\|^2 + 2\|\tilde{z}_t\|^2$).

By setting $\beta_t := \frac{\gamma_t}{1 + \lambda_t r(x_t)}$ and using the expressions $g_t = \frac{\nabla f(x_t)}{1 + \lambda_t r(x_t)}$ and $\tilde{z}_t = \frac{Z_t}{1 + \lambda_t r(x_t)}$, from (C.6), it follows:

$$\begin{aligned}
f(x_{t+1}) - f(x_t) &\le -\beta_t \|\nabla f(x_t)\|^2 - \gamma_t \langle \nabla f(x_t), \tilde{z}_t \rangle + \frac{(L_0 + L_1 \|\nabla f(x_t)\|)}{1 + \lambda_t r(x_t)} \gamma_t \beta_t e^{L_1 \gamma_t (\|g_t\| + \|\tilde{z}_t\|)} \|\nabla f(x_t)\|^2 \\
&\quad + \frac{(L_0 + L_1 \|\nabla f(x_t)\|)}{1 + \lambda_t r(x_t)} \gamma_t^2 e^{L_1 \gamma_t \|g_t\| + \|\tilde{z}_t\|} \frac{\|Z_t\|^2}{1 + \lambda_t r(x_t)} \\
&\le -\beta_t \|\nabla f(x_t)\|^2 - \gamma_t \langle \nabla f(x_t), \tilde{z}_t \rangle + \frac{C_1 \gamma_t}{\lambda_t} e^{L_1 \gamma_t (\|g_t\| + \|\tilde{z}_t\|)} \beta_t \|\nabla f(x_t)\|^2 \\
&\quad + \frac{C_1 \gamma_t^2}{\lambda_t} e^{L_1 \gamma_t \|g_t\| + \|\tilde{z}_t\|} \frac{\|Z_t\|^2}{1 + \lambda_t r(x_t)}
\end{aligned} \tag{C.7}$$

where in the second inequality we used the fact $L_0 + L_1 \|\nabla f(x_t)\| \le \frac{C_1}{\lambda_t}(1 + \lambda_t r(x_t))$ for some suitable constant $C_1$.

By taking expectations conditionally on $\mathcal{F}_t$ from both sides in (C.7), it follows:

$$\begin{aligned}
\mathbb{E}_t[f(x_{t+1})] - f(x_t) &\le -\beta_t \|\nabla f(x_t)\|^2 + \frac{C_1 \gamma_t}{\lambda_t} e^{L_1 \gamma_t \|g_t\|} \beta_t \|\nabla f(x_t)\|^2 \mathbb{E}_t[e^{L_1 \gamma_t \|\tilde{z}_t\|}] \\
&\quad + \frac{C_1 \gamma_t^2}{\lambda_t (1 + \lambda_t r(x_t))} e^{L_1 \gamma_t \|g_t\|} \mathbb{E}_t[e^{L_1 \gamma_t \|\tilde{z}_t\|} \|Z_t\|^2] \\
&\le -\beta_t \|\nabla f(x_t)\|^2 + C_1 e^{C_0 \frac{\gamma_t}{\lambda_t}} \frac{\gamma_t}{\lambda_t} \beta_t \|\nabla f(x_t)\|^2 \mathbb{E}_t[e^{L_1 \gamma_t \|\tilde{z}_t\|}] \\
&\quad + C_1 e^{C_0 \frac{\gamma_t}{\lambda_t}} \frac{\gamma_t^2}{\lambda_t (1 + \lambda_t r(x_t))} \left( \mathbb{E}_t[e^{2L_1 \gamma_t \|\tilde{z}_t\|}] \right)^{\frac{1}{2}} \left( \mathbb{E}_t[\|Z_t\|^4] \right)^{\frac{1}{2}}
\end{aligned} \tag{C.8}$$

where in the second inequality we used the fact $L_1 \|g_t\| \le \frac{C_0}{\lambda_t}$ and the Cauchy-Schwarz inequality for the expectations.

On the one hand, by using Lemma C.1 we have $\mathbb{E}_t[e^{2L_1 \gamma_t \|\tilde{z}_t\|}] \le e^{2c \frac{\gamma_t}{\lambda_t}}$. On the other hand, by using Lemma B.4, with $p = 2$ and (ABC$_\sigma$) of Assumption 4, it holds $\mathbb{E}_t[\|Z_t\|^4] \le c_0 + c_1(f(x_t) - f_*)^2 + c_2\|\nabla f(x_t)\|^4$, for some suitable constants $c_0$, $c_1$ and $c_2$. Therefore, from (C.8), it follows

$$\begin{aligned}
\mathbb{E}_t[f(x_{t+1})] - f(x_t) &\le -\beta_t \|\nabla f(x_t)\|^2 + C_1 e^{C_0' \frac{\gamma_t}{\lambda_t}} \frac{\gamma_t}{\lambda_t} \beta_t \|\nabla f(x_t)\|^2 \\
&\quad + C_1 e^{C_0' \frac{\gamma_t}{\lambda_t}} \frac{\gamma_t^2}{\lambda_t (1 + \lambda_t r(x_t))} \left( c_0 + c_1(f(x_t) - f_*)^2 + c_2 \|\nabla f(x_t)\|^4 \right)^{\frac{1}{2}} \\
&\le -\left( 1 - C_2 \frac{\gamma_t}{\lambda_t} e^{C_0' \frac{\gamma_t}{\lambda_t}} \right) \beta_t \|\nabla f(x_t)\|^2 + C_3 e^{C_0' \frac{\gamma_t}{\lambda_t}} \frac{\gamma_t^2}{\lambda_t} (f(x_t) - f_*) + C_4 e^{C_0' \frac{\gamma_t}{\lambda_t}} \frac{\gamma_t^2}{\lambda_t}
\end{aligned} \tag{C.9}$$

where in the last one inequality the subadditivity of the square root, for some suitable constants $C_0'$, $C_2$, $C_3$ and $C_4$.

Since $\frac{\gamma_t}{\lambda_t} \to 0$, there exists some $\delta \in (0,1)$ for large enough $t$. By subtracting $f_*$ and taking total expectations from both sides in (C.8), it follows:

$$\mathbb{E}[f(x_{t+1}) - f_*] \leq \left(1 - C_3' \frac{\gamma_t^2}{\lambda_t}\right) \mathbb{E}[f(x_t) - f_*] - \delta \beta_t \, \mathbb{E}[\|\nabla f(x_t)\|^2] + C_4' \frac{\gamma_t^2}{\lambda_t} \tag{C.10}$$

for some suitable constants $C_3'$ and $C_4'$.

Since $\frac{\gamma_t^2}{\lambda_t}$ is summable, by using the Robbins-Siegmund lemma [69] (see also [26]), we infer that the limit $\lim_{t \to +\infty} f(x_t)$ exists and is finite a.s. and

$$\sum_{t=0}^{+\infty} \beta_t \|\nabla f(x_t)\|^2 < +\infty \quad \text{a.s.} \tag{C.11}$$

Since $f$ is coercive and converges a.s. and $r(x)$ is coercive, it follows that the sequence $\{x_t\}_{t \geq 0}$, as also $r(x_t)$ are bounded a.s.

Moreover, from the $(L_0\text{-}L_1)$-smoothness (2) and the a.s. boundedness of $\{x_t\}_{t \geq 0}$, with probability 1, there exists some constant $L$, such that for any $h \geq 0, t \geq 0$, it holds

$$\left| \|\nabla f(x_{t+h})\| - \|\nabla f(x_t)\| \right| \leq L\|x_{t+h} - x_t\| = L\|\sum_{s=t}^{t+h} \gamma_s g_s + \gamma_s \tilde{z}_s\| = L\|\sum_{s=t}^{t+h} \beta_s \nabla f(x_s) + \gamma_s \tilde{z}_s\|$$

$$\leq L \sum_{s=t}^{t+h} \beta_s \|\nabla f(x_s)\| + \|\sum_{s=t}^{t+h} \frac{\gamma_s}{1 + \lambda_s r_s} Z_s\| \tag{C.12}$$

The random variable $v_t = \sum_{s=0}^{t} \frac{\gamma_s}{1 + \lambda_s r_s} Z_s$ is martingale whose variance is bounded by

$$\mathbb{E}_t[\|v_t\|^2] \leq \mathbb{E}_t\left[\sum_{s=0}^{t} \gamma_s^2 \|Z_s\|^2\right] = \sum_{s=0}^{t} \gamma_s^2 \, \mathbb{E}_t[\|Z_s\|^2] \leq C_1 \sum_{s=0}^{t} \gamma_s^2 (1 + r(x_s)) \leq C_2 \sum_{s=0}^{t} \gamma_s^2 < +\infty. \tag{C.13}$$

Therefore $v_\infty = \lim_{t \to \infty} v_t = \sum_{t=0}^{\infty} \frac{\gamma_t}{1 + \lambda_t r(x_t)} Z_t$ exists and is a.s. finite (this is an immediate consequence of the martingale convergence theorem). Moreover from (C.11) we have $\sum_{t=0}^{+\infty} \beta_t \|\nabla f(x_t)\|^2 < +\infty$ a.s. and since $\lambda_t$ and $r(x_t)$ are bounded, it follows that $\sum_{t=0}^{\infty} \beta_t = +\infty$. Therefore, by using Lemma B.2 with $a_t = \|\nabla f(x_t)\|$, $\eta_t = \beta_t$ and $v_t = \sum_{s=0}^{t} \frac{\gamma_s}{1 + \lambda_s r_s} Z_s$, we conclude that $\|\nabla f(x_t)\|$ converges to 0 almost surely.

■

# D    Convergence analysis of (DTSGD)

In this section we provide the convergence analysis of Algorithm DTSGD and the associated proof of Theorem 2.

Recall that, for any $\epsilon \in (0,1]$, DTSGD can be expressed as follows:

$$x_{t+1} = -\gamma_t \frac{\hat{g}_t - ax_t}{[1 + \lambda_t r(x_t)^{1/\varepsilon}]^\varepsilon} - \gamma_t a x_t$$
$$= x_k - \gamma_t (g_t + \tilde{z}_t) \tag{D.1}$$

where

$$g_t = \frac{\nabla f(x_t) - ax_t}{\left(1 + \lambda_t r(x_t)^{\frac{1}{\epsilon}}\right)^\epsilon} + ax_t$$

and

$$\tilde{z}_t = \frac{Z_t}{\left(1 + \lambda_t r(x_t)^{\frac{1}{\epsilon}}\right)^\epsilon}$$

Next, we present some useful lemmas that will be employed in the proof of Theorem 2.

**Lemma D.1.** *Let the Assumption 1 and (20) be in force. The tamed term $g_t$ has the following properties:*

•

$$\|g_t\| \leq C(\lambda_t^{-\epsilon} + |x|) \tag{D.2}$$

•

$$\|g_t - \nabla f(x_t)\|^2 \leq C\lambda_t^2 \left(\|\nabla f(x_t)\|^2 + a^2\|x_t\|^2\right) r(x_t)^{\frac{2}{\epsilon}} \tag{D.3}$$

• *There exist $\alpha, b$ such that*

$$\langle g_t, x_t \rangle \geq \alpha\|x_t\|^2 - b \tag{D.4}$$

*Proof.*

$$\|g_t\| = \left\| ax_t + \frac{\nabla f(x_t) - ax_t}{\left(1 + \lambda_t r(x_t)^{\frac{1}{\epsilon}}\right)^{\epsilon}} \right\|$$

$$\leq a\|x_t\| + \frac{\|\nabla f(x_t)\|}{\left(1 + \lambda_t r(x_t)^{\frac{1}{\epsilon}}\right)^{\epsilon}}$$

Noticing that

$$\lambda_t r(x_t)^{\frac{1}{\epsilon}} \leq 1 + \lambda_t r(x_t)^{\frac{1}{\epsilon}} \implies r(x_t)^{\frac{1}{\epsilon}} \leq \lambda_t^{-1}(1 + \lambda_t r(x_t)^{\frac{1}{\epsilon}})$$

$$\implies r(x_t) \leq \lambda_t^{-\epsilon}(1 + \lambda_t r(x_t)^{\frac{1}{\epsilon}})^{\epsilon},$$

it follows that

$$\|\nabla f(x_t)\| \leq C(1 + r(x_t)) \leq C + \lambda_t^{-\epsilon}(1 + \lambda_t r(x_t)^{\frac{1}{\epsilon}})^{\epsilon} \leq (C+1)\lambda_t^{-\epsilon}(1 + \lambda_t r(x_t)^{\frac{1}{\epsilon}})^{\epsilon}$$

which concludes the proof of (D.2).

For the proof of point (D.3), since $t \geq 0$

$$1 - (1+s)^{-\epsilon} = \epsilon s \int_0^1 (1 + s\tau)^{-\epsilon-1} d\tau,$$

it follows that

$$|1 - (1+s)^{-\epsilon}|^2 \leq \epsilon^2 s^2$$

Setting $s := \lambda_t r(x_t)^{1/\epsilon}$, we obtain

$$\|g_t - \nabla f(x_t)\|^2 = \frac{|1 - (1+s)^{-\epsilon}|^2 \|\nabla f(x_t) - ax_t\|^2}{(1+s)^{2\epsilon}}$$

$$\leq C\lambda_t^2 \left(\|\nabla f(x_t)\|^2 + a^2\|x_t\|^2\right) r(x_t)^{\frac{2}{\epsilon}}$$

which completes the proof of (D.3).

Finally, from the definition of $g_t$, it holds

$$\langle g_t, x_t \rangle = a\|x_t\|^2 + \frac{\langle \nabla f(x_t), x_t \rangle - a\|x_t\|^2}{(1 + \lambda_t r(x_t)^{\frac{1}{\epsilon}})^{\epsilon}}$$

Also note that from Assumption 1 there exist $\bar{a}, b > 0$ such that

$$\langle \nabla f(x_t), x_t \rangle \geq \bar{a} - b.$$

In the case

$$\langle \nabla f(x_t), x_t \rangle - a\|x_t\|^2 \leq 0$$

one obtains

$$\frac{\langle \nabla f(x_t), x_t \rangle - a\|x_t\|^2}{(1 + \lambda_t r(x_t)^{\frac{1}{\epsilon}})^\epsilon} \geq \langle \nabla f(x_t), x_t \rangle - a\|x_t\|^2$$

which leads to

$$\begin{aligned} \langle g_t, x_t \rangle &\geq a\|x_t\|^2 + \langle \nabla f(x_t), x_t \rangle - a\|x_t\|^2 \\ &= \langle \nabla f(x_t), x_t \rangle \\ &\geq \bar{a}\|x_t\|^2 - b. \end{aligned}$$

On the other hand, in the case

$$\langle \nabla f(x_t), x_t \rangle - a\|x_t\|^2 \geq 0$$

one has that

$$\langle g_t, x_t \rangle \geq a|x|^2 + 0 = a|x|^2 \geq a\|x_t\|^2 - b.$$

Setting $\alpha := \min\{a, \bar{a}\}$ allows us to conclude the proof of (D.4) and Lemma D.1. ∎

**Lemma D.2.** *Suppose that Assumptions 1–3 hold. Assume further that (DTSGD) is run with a scale function $r(x)$ as per (20)-(21) and stepsize parameters given by (22).*
*Let $\Delta_t = \|x_t - \gamma_t g_t\|$. For $\gamma_t \lambda_t^{-2\epsilon} \leq 1$ and $\gamma_t \leq \frac{1}{4a}$ the holds*

$$\|\Delta_t\|^2 \leq (1 - \gamma_t A)\|x_t\|^2 + \gamma_t B$$

*where $A = \frac{\alpha}{2}$ and $B = C$*

*Proof.* Since

$$\|\Delta_t\|^2 = \|x_t\|^2 - \gamma_t \langle g_t, x_t \rangle + \gamma_t^2 \|g_t\|^2$$

Using (D.4) and (D.2) one obtains

$$\|\Delta_t\|^2 \leq (1 - \alpha\gamma_t)\|x_t\|^2 + C\gamma_t^2 \lambda^{-2\epsilon} + 2\gamma_t^2 a^2 \|x_t\|^2$$

Since $2\gamma_t a^2 \leq \frac{\alpha}{2}$ and $\gamma_t \lambda^{-2\epsilon}$ one obtains

$$\|\Delta_t\|^2 \leq (1 - A\gamma_t)\|x_t\|^2 + \gamma_t B$$

where $A = \frac{\alpha}{2}$ and $B = C$ ∎

**Lemma D.3.** *Let Assumptions 3, (20) and (22) be in force.*
*Let $u$ a deterministic vector. There holds*

$$\mathbb{E}_t[\exp(\tilde{a}\gamma_t \langle u, \tilde{z}_t \rangle)] \leq \exp\left(\frac{\tilde{a}^2 \gamma_t^2 C \|u\|^2}{2\lambda_t^{2\epsilon}}\right)$$

*Proof.* Since $e_t$ is subgaussian conditioned on $x_t$, for any deterministic vector $u$ and any $a_0 > 0$ there holds

$$\mathbb{E}_t[\exp(a_0\gamma_t \langle u, \tilde{z}_t \rangle)] \leq \exp\left(\frac{a_0^2 \gamma_t^2 \sigma^2(x) \|u\|^2}{2}\right)$$

Setting $a_0 = \frac{\tilde{a}}{(1 + \lambda_t r(x_t)^\epsilon)^{\frac{1}{\epsilon}}}$ one obtains

$$\mathbb{E}_t[\exp(\tilde{a}\gamma_t \langle u, \tilde{z}_t \rangle)] \leq \exp\left(\frac{\tilde{a}^2 \gamma_t^2 \sigma^2(x) \|u\|^2}{2(1 + \lambda_t r(x_t)^{\frac{1}{\epsilon}})^\epsilon}\right)$$

Since $\frac{\sigma^2(x)}{(1 + \lambda_t r(x_t)^{\frac{1}{\epsilon}})^{2\epsilon}} \leq C(1 + \lambda_t^{-2\epsilon}) \leq C'\lambda_t^{-2\epsilon}$ one deduces the result. ∎

**Lemma D.4.** *Let Assumptions 3, (20) and (22) be in force. Then, there exists $c, \hat{C}$ universal constants such that*

$$\mathbb{E}_t\left[\exp\left(\frac{c\|Z_t\|^2}{\sigma^2(x)}\right)\right] \leq e^{\hat{C}}$$

*In addition, if $\tilde{a}\frac{\gamma_t C_0}{\lambda_t^{2\epsilon} c} < 1$*

$$\mathbb{E}_t\left[\exp\left(\tilde{a}\gamma_t^2\|\tilde{z}_t\|^2\right)\right] \leq e^{\tilde{a}\gamma_t\hat{C}}$$

*Proof.* The first part of the proof is given in [80, Proposistion 6.2.1]. For the second part, from the definition of $\tilde{z}_t$, we have

$$\mathbb{E}_t\left[\exp\left(\tilde{a}\gamma_t^2\|\tilde{z}_t\|^2\right)\right] = \mathbb{E}_t\left[\exp\left(\frac{c}{\sigma^2(x)}\tilde{a}\gamma_t^2\frac{\sigma^2(x)}{c(1+\lambda_t r(x)^{\frac{1}{\epsilon}})^{\frac{2}{\epsilon}}}\|Z_t\|^2\right)\right].$$

Using the fact that $\frac{\sigma^2(x)}{(1+\lambda_t r(x)^{\frac{1}{\epsilon}})^{\frac{2}{\epsilon}}} \leq C_0\lambda_t^{-2\epsilon}$ for some $C_0 > 0$, it follows

$$\mathbb{E}_t\left[\exp\left(\tilde{a}\gamma_t^2\|\tilde{z}_t\|^2\right)\right] \leq \mathbb{E}_t\left[\exp\left(\frac{c}{\sigma^2(x)}\|Z_t\|^2\right)\right]^{\frac{C_0\tilde{a}\gamma_t^2}{c\lambda_t^{2\epsilon}}} \leq \mathbb{E}_t\left[\exp\left(\frac{c}{\sigma^2(x)}\|Z_t\|^2\right)\right]^{\gamma_t}$$

Then since $\gamma_t < 1$ by hoelder inequality it follows that

$$\mathbb{E}_t\left[\exp\left(\tilde{a}\gamma_t^2\|\tilde{z}_t\|^2\right)\right] \leq \left(\mathbb{E}_t\left[\exp\left(\frac{c}{\sigma^2(x)}\|Z_t\|^2\right)\right]\right)^{\tilde{a}\gamma_t} \leq e^{\tilde{a}\hat{C}\gamma_t}$$

$\blacksquare$

**Lemma D.5.** *Let Assumptions 3, (20), (22) be in force. There holds*

$$\mathbb{E}_t\left[e^{2L_1\gamma_t\|\tilde{z}_t\|}\right] \leq e^{C'\gamma_t\lambda^{-\epsilon}}$$

*and*

$$\mathbb{E}_t\left[\|\tilde{z}_t\|^{2p}\right] \leq \frac{C}{\lambda^{2p\epsilon}}$$

*Proof.* For the first part, applying Lemma B.3 for $\eta = \frac{2L_1\gamma_t}{(1+\lambda_t r(x_t)^{\frac{1}{\epsilon}})^{\epsilon}}$ since

$$\sigma(x_t)\eta \leq C\frac{1+r(x_t)}{(1+\lambda_t r(x_t)^{\frac{1}{\epsilon}})^{\epsilon}} \leq C\gamma_t\gamma_t^{-\epsilon} \leq C$$

yields the result. The second part follows immediately from Lemma B.4 as

$$\mathbb{E}_t\left[\|\tilde{z}_t\|^4\right] = (1+\lambda_t r(x_t)^{\frac{1}{\epsilon}})^{-4\epsilon}\mathbb{E}_t\left[\|Z_t\|^4\right] \leq (1+\lambda_t r(x_t)^{\frac{1}{\epsilon}})^{-4\epsilon}C\sigma(x_t)^4 \leq C\lambda^{-4\epsilon}$$

$\blacksquare$

**Lemma D.6.** *Suppose that Assumptions 1–3 hold. Assume further that (DTSGD) is run with a scale function $r(x)$ as per (20)-(21) and stepsize parameters given by (22). There exist $\bar{a} > 0$ such that*

$$\sup_t \mathbb{E}\left[\exp(\bar{a}\|x_t\|^2)\right] < \infty$$

*Consequently*

$$\sup_t \mathbb{E}[\exp(c\|x_t\|)] < \infty \quad \forall c > 0.$$

*Proof.* By the assumption, $Z_t$ is subGaussian with parameter $\sigma(x) = C(1 + r(x))$ By Lemma D.3 for $\tilde{a} > 0$, for every deterministic vector $u$,

$$\mathbb{E}[\exp(a\gamma_t \langle u, \tilde{z}_t \rangle)] \leq \exp\left(0.5\tilde{a}^2\gamma_t^2 \frac{C}{\lambda_t^{2\epsilon}} \|u\|^2\right) \tag{D.5}$$

and by Lemma D.4, there exists $c > 0$ such that

$$\mathbb{E}\left[\exp\left(\frac{c}{\sigma^2(x_t)} \|Z_t\|^2\right)\right] \leq e^{\hat{C}}$$

which also implies that if $\tilde{a}\frac{\gamma_t}{\lambda_t^2} < c$

$$\mathbb{E}\left[\exp\left(\tilde{a}\gamma_t^2 \|\tilde{z}_t\|^2\right)\right] \leq e^{\tilde{a}\gamma_t \hat{C}} \tag{D.6}$$

Let $\tilde{a} > 0$. and set $V_{\tilde{a}}(x_t) = \mathbb{E}\left[\exp(\tilde{a}\|x_t\|^2)\right]$ and $\Delta_t = x_t - \gamma_t g_t$.

$$\begin{aligned}
\mathbb{E}_t\left[\exp \tilde{a}\|x_{t+1}\|^2\right] &= \mathbb{E}_t\left[\exp\left(\tilde{a}\|\Delta_t + \gamma_t \tilde{z}_t\|^2\right)\right] \\
&= \mathbb{E}_t\left[\exp\left(\tilde{a}\left(\|\Delta_t\|^2 + 2\gamma_t \langle \Delta_t, \tilde{z}_t \rangle\right) + \gamma_t^2\|\tilde{z}_t\|^2\right)\right] \\
&\leq e^{\tilde{a}\|\Delta_t\|^2} \mathbb{E}_t\left[e^{\tilde{a}\left(2\gamma_t \langle \Delta_t, \tilde{z}_t \rangle + \gamma_t^2\|\tilde{z}_t\|^2\right)}\right]
\end{aligned} \tag{D.7}$$

Using Lemma D.2 yields

$$\|\Delta_t\|^2 \leq (1 - \gamma_t A)\|x_t\|^2 + \gamma_t B. \tag{D.8}$$

which leads to

$$\mathbb{E}_t\left[e^{\tilde{a}\|\Delta_t\|^2}\right] \leq \mathbb{E}_t\left[e^{\tilde{a}(1-\gamma_t A)\|x_t\|^2}\right] e^{\tilde{a}\gamma_t B} \tag{D.9}$$

Applying (D.5) for $u = \|\Delta_t\|$ (which is treated as deterministic inside conditiondial expectation with respect to $x_t$), yields

$$\mathbb{E}_t\left[e^{\tilde{a}(2\gamma_t \langle \Delta_t, \tilde{z}_t \rangle)}\right] \leq \exp\left(\frac{\tilde{a}^2}{2} \frac{\gamma_t^2 c_0}{\lambda_t^{2\epsilon}} \|\Delta_t\|^2\right) \tag{D.10}$$

Assuming that $\bar{a}^2 \frac{c_0 \gamma_t}{\lambda_k^{2\epsilon}} \leq \frac{A}{4}$ applying D.8 yields

$$\mathbb{E}_t\left[e^{\tilde{a}(2\gamma_t \langle \Delta_t, \tilde{z}_t \rangle)}\right] \leq e^{\tilde{a}\frac{A}{4}\gamma_t\|\Delta_t\|^2} \leq e^{\gamma_t \frac{A}{4}\|x_t\|^2 + \gamma_t^2 \tilde{a}\frac{A}{4}B} \tag{D.11}$$

Substituting (D.9),(D.11),(D.6) into (D.7) yields

$$\mathbb{E}_t\left[e^{\tilde{a}\|x_{t+1}\|^2}\right] \leq e^{\tilde{a}\left((1-\gamma_t \frac{A}{2})\|x_t\|^2 + \gamma_t(B+\hat{C})\right)}.$$

Let $R = \sqrt{4\frac{B+\hat{C}}{A}}$. In the event that $\|x_t\| \geq R$

$$\mathbb{E}_t\left[e^{\tilde{a}\|x_{t+1}\|^2}\right] \leq e^{\tilde{a}\|x_t\|^2} e^{-\gamma_t \tilde{a}\frac{R^2 A}{4}} \tag{D.12}$$

while for $\|x_t\| \leq R$

$$\mathbb{E}_t\left[e^{\tilde{a}\|x_{t+1}\|^2}\right] \leq e^{\tilde{a}\|x_t\|^2} e^{\gamma_t(B+\hat{C})} = e^{\tilde{a}\|x_t\|^2} e^{-\gamma_t \frac{R^2 A}{4}} + e^{\tilde{a}\|x_t\|^2}\left(e^{\gamma_t(B+\hat{C})} - e^{-\gamma_t \frac{R^2 A}{4}}\right)$$

Using that

$$e^{\gamma_t(B+\hat{C})} - e^{-\gamma_t \frac{R^2 A}{4}} = e^{\gamma_t(B+\hat{C})}(1 - e^{-\gamma_t(B+\hat{C}+\frac{R^2 A}{4})} \leq \gamma_t e^{\gamma_t(B+\hat{C})}(B + \hat{C} + \frac{R^2 A}{4})$$

one obtains that for $\|x_t\| \leq R$

$$\mathbb{E}_t\left[e^{\tilde{a}\|x_{t+1}\|^2}\right] \leq e^{\tilde{a}\|x_t\|^2} e^{-\gamma_t \frac{R^2 A}{4}} + \gamma_t e^{\gamma_t(B+\hat{C})}(B + \hat{C} + \frac{R^2 A}{4})e^{\tilde{a}R^2}. \tag{D.13}$$

Combining (D.12) and (D.13) and taking total expectations, we obtain

$$\mathbb{E}\left[e^{\tilde{a}\|x_{t+1}\|^2}\right] \leq \mathbb{E}\left[e^{\tilde{a}\|x_t\|^2}\right]e^{-\gamma_t R^2/4} + \gamma_t b \tag{D.14}$$

for some constant $b$. By an induction argument, from (D.14), it follows

$$\begin{aligned}
\mathbb{E}[e^{\tilde{a}\|x_{t+1}\|^2}] &\leq \mathbb{E}[e^{\tilde{a}\|x_0\|^2}]\prod_{k=0}^{t}e^{-b\gamma_k} + b\sum_{m=0}^{t}\gamma_m\prod_{k=m+1}^{t}e^{-b\gamma_k} \\
&= \mathbb{E}[e^{\tilde{a}\|x_0\|^2}]e^{-bS_t} + b\sum_{m=0}^{t}\gamma_m e^{-b(S_t - S_m)} \\
&\leq \mathbb{E}[e^{\tilde{a}\|x_0\|^2}] + be^{-bS_t}\sum_{m=0}^{t}\gamma_m e^{bS_m}
\end{aligned} \tag{D.15}$$

where we used the notation $S_t = \sum_{j=0}^{t}\gamma_j$.

Moreover, since $\gamma_t$ decays to 0, there exists some constant $\Gamma_0$, such that

$$\gamma_m e^{bS_m} = \frac{\gamma_m}{1 - e^{-bS_m}}\left(e^{bS_m} - e^{bS_{m-1}}\right) \leq \Gamma_0\left(e^{bS_m} - e^{bS_{m-1}}\right) \tag{D.16}$$

By injecting (D.16) into (D.15), we find:

$$\mathbb{E}[e^{\tilde{a}\|x_{t+1}\|^2}] \leq \mathbb{E}[e^{\tilde{a}\|x_0\|^2}] + \Gamma_0 b \tag{D.17}$$

which allows to conclude the proof.

∎

We are now ready to proceed with the proof of Theorem 2

**Theorem 2.** *Suppose that Assumptions 1–3 hold. Assume further that (DTSGD) is run with a scale function $r(x)$ as above, and step-size / taming parameters such that*

$$\sum_{t=1}^{\infty}\gamma_t = \infty, \quad \sum_{t=1}^{\infty}\gamma_t\lambda_t < \infty \quad and \quad \sum_{t=1}^{\infty}\frac{\gamma_t^2}{\lambda_t^{2\varepsilon}} < \infty. \tag{22}$$

*Then, with probability 1, we have $\lim_{t\to\infty}\nabla f(x_t) = 0$. In particular, if $\gamma_t \propto 1/t^p$ and $\lambda_t \propto 1/t^q$, then $x_t$ converges to* crit $f$ *(a.s.) as long as $0 \leq 1 - p < q < (p - 1/2)/\varepsilon$.*

***Proof of Theorem 2.*** In the same way as in (C.6) in the proof of Theorem 1, applying (B.1) in Lemma B.1 with $x' = x_{t+1}$ and $x = x_t$, we find:

$$f(x_{t+1}) - f(x_t) \leq -\gamma_t\langle\nabla f(x_t), g_t\rangle - \gamma_t\langle\nabla f(x_t), \tilde{z}_t\rangle + (L_0 + L_1\|\nabla f(x_t)\|)e^{L_1\gamma_t\|g_t\| + \|\tilde{z}_t\|}\gamma_t^2\left(\|g_t\|^2 + \|\tilde{z}_t\|^2\right) \tag{D.18}$$

The first term in (D.18) can be bounded as follows:

$$\begin{aligned}
-\gamma_t\langle\nabla f(x_t), g_t\rangle &= -\gamma_t\|\nabla f(x_t)\|^2 - \gamma_t\langle\nabla f(x_t), g_t - \nabla f(x_t)\rangle \\
&\leq -\gamma_t\|\nabla f(x_t)\|^2 + \frac{\gamma_t}{2}\|\nabla f(x_t)\|^2 + 2\gamma_t\|g_t - \nabla f(x_t)\|^2 \\
&= -\frac{\gamma_t}{2}\|\nabla f(x_t)\|^2 + 2\gamma_t\|g_t - \nabla f(x_t)\|^2 \\
&\leq -\frac{\gamma_t}{2}\|\nabla f(x_t)\|^2 + 2C\gamma_t\lambda_t^2(1 + r(x_t)^{2/\epsilon})(\|\nabla f(x_t)\|^2 + a^2\|x_t\|^2)
\end{aligned} \tag{D.19}$$

while the last inequality follows from (D.3) in Lemma D.1. Notice that taking conditional expectation in the third term one deduces that for $\phi_1(x_t) = (L_0 + L_1\|\nabla f(x_t)\|)e^{L_1\gamma_t\|g_t\|}\|g_t\|^2$ and $\phi_2 = (L_0 + L_1\|\nabla f(x_t)\|)e^{L_1\gamma_t\|g_t\|}$

$$\begin{aligned}
&\mathbb{E}_t\left[(L_0 + L_1\|\nabla f(x_t)\|)e^{L_1\gamma_t(\|g_t\| + \|\tilde{z}_t\|)}\gamma_t^2\left(\|g_t\|^2 + \|\tilde{z}_t\|^2\right)\right] \\
&= \gamma_t^2\,\mathbb{E}_t\left[e^{L_1\gamma_t\tilde{z}_t}\right]\phi_1(x_t) + \gamma_t^2\,\mathbb{E}_t\left[e^{L_1\gamma_t\tilde{z}_t}\|\tilde{z}_t\|^2\right]\phi_2(x_t) \\
&\leq \gamma_t^2\,\mathbb{E}_t\left[e^{L_1\gamma_t\|\tilde{z}_t\|}\right]\phi_1(x_t) + \gamma_t^2\sqrt{\mathbb{E}_t\left[e^{2L_1\gamma_t\|\tilde{z}_t\|}\right]}\sqrt{\mathbb{E}_t\left[\|\tilde{z}_t\|^4\right]}\phi_2(x_t)
\end{aligned}$$

Noticing that by Lemma D.5 $\mathbb{E}_t\left[\|\tilde{z}_t\|^4\right] \le \frac{c}{\lambda_t^{4\epsilon}}$ and

$$\mathbb{E}_t\left[e^{2L_1\gamma_t\|\tilde{z}_t\|}\right] \le e^{C'\frac{\gamma_t}{\lambda_t^\epsilon}} \le C$$

and using the fact that since $\|g_t\| \le a\|x_t\| + C\lambda^{-\epsilon}$ it is easy to see that

$$\phi_1(x_t) \le (L_0 + L_1\nabla f(x_t))e^{CL_1\gamma_t\lambda_t^{-\epsilon}+L_1\gamma_t a\|x_t\|}(2a^2\|x_t\|^2 + \lambda_t^{-2\epsilon}) \le Ce^{a_0\|x_t\|}(2a\|x_t\|^2 + \lambda_t^{-2\epsilon})$$

for some $a_0 > 0$ and

$$\phi_2(x_t) \le L_0 + L_1\nabla f(x_t))e^{CL_1\gamma_t\lambda_t^{-\epsilon}+L_1\gamma_t a\|x_t\|} \le C(1 + e^{a_1\|x_t\|})$$

for some $a_1 > 0$. ringing all together

$$\mathbb{E}_t\left[(L_0 + L_1\|\nabla f(x_t)\|)e^{L_1\gamma_t(\|g_t\|+\|\tilde{z}_t\|)}\gamma_t^2\left(\|g_t\|^2 + \|\tilde{z}_t\|^2\right)\right] \le \frac{\gamma_t^2}{\lambda_t^{2\epsilon}}C(1 + e^{c\|x_t\|}) \tag{D.20}$$

Taking expectations in (D.18) and combining (D.19) and (D.20) yields

$$\mathbb{E}_t[f(x_{t+1})] \le f(x_t) - \frac{\gamma_t}{2}\|\nabla f(x_t)\|^2 + \gamma_t\lambda_t^2(1 + r(x_t)^{2/\epsilon})(|\nabla f(x_t)|^2 + a^2\|x_t\|^2) + C\frac{\gamma_t^2}{\lambda_t^{2\epsilon}}(1 + e^{c\|x_t\|}).$$

Since $r(x_t), \nabla f(x_t)$ have at most exponential growth and we are able to control exponential square moments from Lemma D.6, by the summability conditions on the noise we can apply Robbins-Siegmund Lemma to conclude that $f(x_t)$ converges, as also that

$$\sum_t \gamma_t \mathbb{E}\left[\|\nabla f(x_t)\|^2\right] < \infty. \tag{D.21}$$

Since $f$ is coercive and converges almost, surely, it follows that the iterates are bounded a.s, so with probability 1, using the $(L_0\text{-}L_1)$ condition (2), we have

$$\left|\|\nabla f(x_{t+h})\| - \|\nabla f(x_t)\|\right| \le C(1 + e^{c\sup_t\|x_t\|})\|x_{t+h} - x_t\|$$

$$\le C(1 + e^{c\sup_t\|x_t\|})\left\|\sum_{s=t}^{t+h}\gamma_t\frac{\nabla f(x_s)}{\left(1 + \lambda_s r(x_s)^{\frac{1}{\epsilon}}\right)^\epsilon} + \gamma_t A_s\right\| \tag{D.22}$$

$$\le C(1 + e^{c\sup\|x_s\|})\left(\sum_{s=t}^{t+h}\gamma_t\|\nabla f(x_s)\| + \left\|\sum\gamma_t A_s\right\|\right)$$

where we set $A_t = ax_t\left(1 - \frac{1}{\left(1+\lambda_t r(x_t)^{\frac{1}{\epsilon}}\right)^\epsilon}\right) + \tilde{z}_t$. Since

$$\left\|\sum\gamma_t A_t\right\| \le \left\|\sum\gamma_t\tilde{z}_t\right\| + e^{c'\sup_t\|x_t\|}\sum\gamma_t\lambda_t$$

and $\gamma_t\tilde{e}_t$ is a martingale difference with finite variance (therefore converges a.s, so it a.s finite) and $\sum_{t=0}^{\infty}\gamma_t\lambda_t < \infty$, it follows that

$$\left\|\sum\gamma_t A_t\right\| < \infty \quad a.s \tag{D.23}$$

In addition from (D.21) we also have that $\sum_{t=0}^{\infty}\gamma_t\mathbb{E}\left[\|\nabla f(x_t)\|^2\right] < \infty$, therefore

$$\sum_{t=0}^{\infty}\gamma_t\|\nabla f(x_t)\|^2 < \infty \quad a.s$$

Combining this estimate together with (D.22) and using Lemma B.2, with probability 1 we obtain

$$\lim_{t\to\infty}\|\nabla f(x_t)\| = 0 \tag{D.24}$$

which concludes the proof of Theorem 2 $\blacksquare$

# E    Avoidance of saddle points

As we stated in the main body of the paper, the proof of Theorem 3 will require two different threads of arguments: *a*) a series of probabilistic estimates to show that a certain class of stochastic processes avoids zero; and *b*) the construction of a suitable (average) Lyapunov function that grows exponentially along the unstable directions of a strict saddle manifold.

As we mentioned earlier, the basic element of our approach is the observation that both (TSGD) and (DTSGD) can be recast as generalized Robbins–Monro schemes of the form

$$x_{t+1} = x_t - \gamma_t(\nabla f(x_t) + Z_t + b_t) \tag{RM}$$

Concretely, we have:

1. For (TSGD), letting $r_t = \lambda_t r(x_t)$, we get

$$x_{t+1} = x_t - \frac{\gamma_t \hat{g}_t}{1 + r_t} = x_t - \gamma_t \hat{g}_t + \gamma_t \frac{r_t \hat{g}_t}{1 + r_t} \tag{E.1a}$$

   so we obtain (RM) with $b_t \leftarrow r_t \hat{g}_t / (1 + r_t)$.

2. For (DTSGD), letting $\beta_t = [1 + \lambda_t r(x_t)^{1/\varepsilon}]^\varepsilon$ and $a_t$ defined by (19), we get

$$x_{t+1} = x_t - \gamma_t \frac{\hat{g}_t - a x_t}{[1 + \lambda_t r(x_t)^{1/\varepsilon}]^\varepsilon} - \gamma_t a_t x_t = x_t - \gamma_t \hat{g}_t - \gamma_t \frac{\beta_t - 1}{\beta_t} \hat{g}_t - \gamma_t a_t \tag{E.1b}$$

   so we obtain (RM) with $b_t \leftarrow (1/\beta_t - 1)[\hat{g}_t - a x_t]$.

Now, under the assumption $\lambda_t \to 0$ (i.e., $q > 0$), we get that $b_t = \mathcal{O}(\lambda_t)$ whenever $x_t$ lies in some sufficiently large compact set $\mathcal{K}$. In this regard, Theorem 3 will follow from the following more general theorem for generalized Robbins–Monro processes:

**Theorem E.1.** *Let $x_t$ be the sequence of iterates generated by (RM) with step-size and bias parameters of the form $\gamma_t \propto 1/t^p$ and $\|b_t\| = \mathcal{O}(1/t^q)$, with $p \in (1/2, 1]$, $q \in (0, 1]$, and $p/2 < q \leq 1$. If $\mathcal{Q}$ is a strict saddle manifold of $f$, we have $\mathbb{P}(x_t \to \mathcal{Q} \text{ as } t \to \infty) = 0$.*

This theorem is an extension of Theorem 9.1 of Benaïm [5], and involves the addition of the extra bias term $b_t$ and the more relaxed conditions on the saddle manifold $\mathcal{Q}$ (which is not assumed to be a priori hyperbolic in our case). The heavy lifting (and the overall structure of the proof) follows Benaïm [5], with an additional finer handling of the center-stable manifold of $\mathcal{Q}$ building on an argument of Mertikopoulos et al. [59]. The specific introduction of the bias term $b_t$ will subsequently appear in the verification of the probabilistic estimates derived in the following section. For completeness, we provide all the relevant details below, stressing throughout that the analysis follows the proof scheme of Benaïm [5] and Benaïm & Hirsch [6].

**E.1. Probabilistic estimates.**    The stochastic estimates required for our analysis originate in the work of Pemantle [64] and pertain to a specific family of stochastic processes, which we describe below. Concretely, let $Y_t$, $t = 1, 2, \ldots$, be a sequence of random variables that are measurable with respect to $\mathcal{F}_t$, define the cumulative process $V_t = \sum_{s=1}^{t} Y_s$, and suppose that the following inequality holds:

$$\mathbb{E}[V_{t+1}^2 - V_t^2 \mid \mathcal{F}_t] \geq C/t^{2p} \quad \text{for some } C > 0 \text{ and all } t = 1, 2, \ldots \tag{E.2}$$

In this setting, $V_t$ will serve as a quantitative proxy for the "distance" to the set of saddle points $\mathcal{Q}$. At an intuitive level, condition (E.2) asserts that $V_t$ exhibits a growth of order $\Theta(\gamma_t)$ in root-mean-square terms, where the step-size $\gamma_t$ of (SGD) scales as $\gamma_t \propto 1/t^p$. The construction of such processes will be carried out via geometric arguments in the next section. For the time being, we record—without proof—several results ensuring that $V_t$ cannot converge to zero.

**Lemma E.1** (Pemantle [65], Lemma 5.5). *Assume that (E.2) is satisfied for some $p \in (1/2, 1]$. In addition, suppose there exist constants $a, b > 0$ such that, for every $t = 1, 2, \ldots$, the following conditions hold:*

*1. $|Y_t| \leq a/t^p$ almost surely.*

*2. $\mathbb{1}\{V_t > b/t^p\} \mathbb{E}[Y_{t+1} \mid \mathcal{F}_t] \geq 0$ almost surely.*

*Then* $\mathbb{P}(\lim_{t\to\infty} V_t = 0) = 0.$

An earlier version of Lemma E.1 was established by Pemantle [64] in the particular case $p = 1$; however, the underlying arguments extend without essential modification to the entire range $1/2 < p \leq 1$. For a refinement of these estimates that will not be required here, we refer the reader to Benaïm [5, Lemma 9.6].

**E.2. Center manifold theory and geometric constructions.** We now turn to the geometric component of the argument and construct an appropriate Lyapunov function that will enable the application of Lemma E.1. Our approach follows the general blueprint developed by Benaïm & Hirsch [6] and Benaïm [5], and makes essential use of tools from center manifold theory; for background material, we refer the reader to Lee [44], Shub [74] and Robinson [70].

Let $\mathcal{Q}$ denote a strict saddle manifold as introduced in Section 5. For each $q \in \mathcal{Q}$, we decompose the ambient space according to the spectral structure of the Hessian $H(q) = \nabla^2 f(q)$ of $f$ at $q$. Specifically, we define the *center*, *stable*, and *unstable* directions at $q$ as the eigenspaces associated respectively with zero, positive, and negative eigenvalues of $H(q)$, namely:

$$\mathcal{E}_q^c = \{v \in \mathbb{R}^d : H(q)v = 0\} = \ker H(q), \tag{E.3a}$$

$$\mathcal{E}_q^s = \{v \in \mathbb{R}^d : H(q)v = \lambda v \text{ for some } \lambda > 0\}, \tag{E.3b}$$

$$\mathcal{E}_q^u = \{v \in \mathbb{R}^d : H(q)v = \lambda v \text{ for some } \lambda < 0\}, \tag{E.3c}$$

This nomenclature reflects the fact that $H(x) = \mathrm{Jac}(\nabla f(x))$, so these subspaces correspond to directions that are respectively neutral (or *slow*), contracting, and expanding under the gradient flow of $f$, viz.

$$\dot{x}(t) = -\nabla f(x(t)) \tag{GF}$$

More precisely, the center manifold theorem [70, 74] guarantees the existence of a neighborhood $\mathcal{U}$ of $\mathcal{Q}$ together with a submanifold $\mathcal{M} \subseteq \mathcal{V}$, referred to as the *center stable manifold* of $\mathcal{Q}$, such that: *a)* $\mathcal{M}$ is *locally invariant* under the flow $\Phi$, in the sense that there exists $t_0 > 0$ for which $\Phi_t(\mathcal{U} \cup \mathcal{M}) \subseteq \mathcal{M}$ holds for all $t \geq t_0$; and *b)* for every $q \in \mathcal{Q}$, the ambient space admits the decomposition $\mathcal{V} = T_q\mathcal{M} \oplus \mathcal{E}_q^u$, where $T_q\mathcal{M}$ denotes the tangent space of $\mathcal{M}$ at $q$. As a consequence:

*a)* Perturbations in central directions are tangent to $\mathcal{M}$ and therefore evolve "along" $\mathcal{M}$ under (GF);

*b)* Perturbations in stable directions $\mathcal{E}_q^s$ are attracted toward $\mathcal{Q}$ along $\mathcal{M}$ under (GF); and

*c)* Perturbations in unstable directions $\mathcal{E}_q^u$ are transverse to $\mathcal{M}$ and are expelled from a neighborhood of $\mathcal{Q}$ at a linear rate.

An important structural property of $\mathcal{M}$ is that any globally bounded trajectory of (GF) which remains sufficiently close to some $q \in \mathcal{Q}$ must lie entirely within $\mathcal{M}$ [74]. Furthermore, since $\mathcal{Q}$ is assumed to be non-minimal, we have $d_u \equiv \dim \mathcal{E}_q^u \geq 1$, implying that $\mathcal{M}$ has dimension at most $d - 1$. This observation suggests that any perturbation with a nonzero component transverse to $\mathcal{M}$ will be repelled under the dynamics (GF); the following lemma, originally due to Benaïm [5], with an adaptation of an argument by Mertikopoulos et al. [59], formalizes this intuition.

**Lemma E.2.** *Let* $\Psi_t(x) = \nabla_x \Phi_t(q)$ *denote the linearization of the flow associated with* (GF). *Then the following statements hold:*

*1. For every $q \in \mathcal{Q}$, the unstable subspace $\mathcal{E}_q^u$ is preserved by the flow of (GF); more precisely, $\Psi_t(q)\mathcal{E}_q^u = \mathcal{E}_q^u$ for all $t \geq 0$.*

*2. There exists a constant $c > 0$ such that, for all $q \in \mathcal{Q}$, all $w \in \mathcal{E}_q^u$, and all $t \geq 0$, we have*

$$\|\Psi_t(q)w\| \geq e^{ct}\|w\|. \tag{E.4}$$

*Remark* 6. Here and throughout, if $A: V \to V'$ is a linear map between vector spaces and $W \leq V$ is a subspace, we write $AW$ for the image of $W$ under $A$, i.e., $AW \equiv \mathrm{im}_A(W) = \{Aw : w \in W\}$. We will freely identify linear operators with their matrix representations when no ambiguity arises.

*Remark* 7. The argument establishing Lemma E.2—as well as the remainder of the analysis in this section—does not fundamentally rely on the uniform bound $\min \lambda_+(H(q)) \geq c_+$ for the positive eigenvalues of the Hessian (when such eigenvalues are present). This assumption is introduced solely to streamline the exposition and to avoid technicalities related to changes in the dimension of $\mathcal{E}_q^s$; if necessary, the analysis could instead be restricted to subsets of $\mathcal{Q}$ on which this dimension remains constant.

In informal terms, Lemma E.2 asserts that: *a)* the collection of unstable directions is invariant under the gradient flow (GF); and *b)* any perturbation with a component in an unstable direction is exponentially amplified. We now give the proof.

*Proof of Lemma E.2.* Recall that for any $t \geq 0$ and any $x \in \mathcal{X}$, the Jacobian of the flow admits the representation

$$\Psi_t(x) = \nabla_x \Phi_t(x) = \exp(t \operatorname{Jac}(-\nabla f(x))) = \exp(-tH(x)).$$

Since every $q \in \mathcal{Q}$ is a stationary point of (GF), it follows immediately that

$$\begin{aligned}
\Psi_t(q)\mathcal{E}_q^u &= e^{-tH(q)}\mathcal{E}_q^u \\
&= \sum_{k=0}^{\infty} \frac{(-t)^k}{k!}H(q)^k \mathcal{E}_q^u = \sum_{k=0}^{\infty} \frac{(-t)^k}{k!}\mathcal{E}_q^u && \triangleright \text{ since } H(q)\mathcal{E}_q^u = \mathcal{E}_q^u \\
&= e^{-t}\mathcal{E}_q^u = \mathcal{E}_q^u,
\end{aligned} \tag{E.5}$$

which establishes the first claim.

To prove the second claim, let $\{u_i : i = 1, \ldots, d\}$ be an orthonormal basis of eigenvectors of $H(q)$, which exists because $H(q)$ is symmetric. Let $\lambda_i \equiv \lambda_i(q) < 0$ denote the eigenvalue associated with $u_i$, and assume without loss of generality that the indices are ordered so that $\lambda_1 \leq \cdots \leq \lambda_d$. Then $\{u_i : i = 1, \ldots, d_u\}$, where $d_u \equiv \dim \mathcal{E}_q^u$, forms an orthonormal basis of $\mathcal{E}_q^u$.

Fix $w \in \mathcal{E}_q^u$ and write $w = \sum_{i=1}^{d_u} w_i u_i$ in this basis. Using the spectral decomposition of $e^{-tH(q)}$, we obtain

$$\Psi_t(q)w = e^{-tH(q)}w = \sum_{i=1}^{d_u} w_i e^{-t\lambda_i} u_i. \tag{E.6}$$

By orthonormality, this yields

$$\|\Psi_t(q)w\|^2 = \sum_{i=1}^{d_u} e^{-2t\lambda_i} w_i^2 \geq e^{2c_- t}\|w\|^2, \tag{E.7}$$

where $c_- > 0$ is the constant defined in Section 5. This completes the proof. ∎

To continue, we introduce a convenient "projection" from a neighborhood of $\mathcal{Q}$ onto the center stable manifold $\mathcal{M}$. This construction is somewhat indirect, but it will allow us to preserve the unstable geometry induced by (GF).

We begin by considering the collection of unstable directions along $\mathcal{Q}$, viewed as the vector bundle

$$\mathcal{E}_{\mathcal{Q}}^u \equiv \{(q,w) : q \in \mathcal{Q}, \ w \in \mathcal{E}_q^u\}. \tag{E.8}$$

Since each fiber $\mathcal{E}_q^u$ is a linear subspace of $\mathcal{X}$, the assignment $q \mapsto \mathcal{E}_q^u$ may be interpreted as a map from $\mathcal{Q}$ into the Grassmannian $\mathbf{Gr}(d_u, d)$ of $d_u$-dimensional subspaces of $\mathbb{R}^d$. By the Whitney embedding theorem [44], $\mathbf{Gr}(d_u, d)$ admits a smooth embedding as a $d_u(d - d_u)$-dimensional submanifold of $\mathbb{R}^{2d_u(d-d_u)}$. Consequently, $\mathcal{E}_{\mathcal{Q}}^u$ can be viewed as a map $\mathcal{Q} \to \mathbb{R}^{2d_u(d-d_u)}$ whose image lies in $\mathbf{Gr}(d_u, d) \hookrightarrow \mathbb{R}^{2d_u(d-d_u)}$.

Since $\mathcal{Q}$ is closed—as it forms a connected component of $\mathcal{X}^*$—the Tietze extension theorem [3] ensures that this map admits a continuous extension $\pi \colon \mathbb{R}^d \to \mathbb{R}^{2d_u(d-d_u)}$ defined on the entire ambient space. By convolving $\pi$ with a smooth approximate identity supported on $\mathcal{Q}$, we may further assume that this extension is smooth in a neighborhood of $\mathcal{Q}$. In addition, standard results from differential topology [32, Chap. 4] guarantee the existence of a smooth retraction from a neighborhood of $\mathbf{Gr}(d_u, d)$ in $\mathbb{R}^{2d_u(d-d_u)}$ onto $\mathbf{Gr}(d_u, d)$ itself. Composing $\pi$ with such a retraction yields a smooth vector bundle

$$\mathcal{E}_{\mathcal{U}}^u \equiv \{(x,w) : x \in \mathcal{U}, \ w \in \mathcal{E}_x^u\}, \tag{E.9}$$

which coincides with $\mathcal{E}_{\mathcal{Q}}^u$ when restricted to $\mathcal{Q}$ (hence the slight abuse of notation).

By shrinking $\mathcal{U}$ if necessary, we may assume that it is compact and that it agrees with the neighborhood appearing in the definition of $\mathcal{M}$, i.e., $\Phi_t(\mathcal{U} \cap \mathcal{M}) \subseteq \mathcal{M}$ for all sufficiently small $t$. We now define a "projection" from a (possibly smaller) neighborhood of $\mathcal{M}$ onto $\mathcal{M}$.

To this end, consider the mapping $Q \colon \mathcal{E}_{\mathcal{U}}^{u} \to \mathcal{X} \equiv \mathbb{R}^d$ given by vector addition,

$$Q(x, w) = x + w.$$

The zero section $(x, 0)$ of $\mathcal{E}_{\mathcal{U}}^{u}$ is mapped diffeomorphically onto $\mathcal{U}$ by $Q$; therefore, by the inverse function theorem [44], $Q$ is a local diffeomorphism. Let $\mathcal{U}'$ be a neighborhood of $\mathcal{M}$ on which $Q$ is a diffeomorphism, and set $\mathcal{U}_0 = Q(\mathcal{U}')$. This induces a map $\Pi \colon \mathcal{U}_0 \to \mathcal{M}$ characterized by the relation

$$\Pi(y) = x \quad \Longleftrightarrow \quad Q(x, w) = x + w = y. \tag{E.10}$$

The motivation for this construction—rather than, say, a Euclidean projection onto $\mathcal{M}$—is that $\Pi$ is compatible with the unstable geometry generated by (GF). More precisely, the following holds.

**Lemma E.3.** *For $x \in \mathcal{U}$, let $P_x \colon T_x\mathcal{M} \oplus \mathcal{E}_x^u \to T_x\mathcal{M}$ denote the linear projection defined by*

$$\underset{\underset{T_x\mathcal{M}\oplus\mathcal{E}_x^u}{\cap}}{z + w} \quad \mapsto P_x(z + w) = \underset{\underset{T_x\mathcal{M}}{\cap}}{z}\,. \tag{E.11}$$

*Then, for every $x \in \mathcal{U}_0 \cap \mathcal{M}$, we have $\mathrm{Jac}(\Pi(x)) = P_x$.*

*Proof.* Let $y(t)$, $t \in (-1, 1)$, be a smooth curve in $\mathcal{U}_0$ satisfying $y(0) = x \in \mathcal{M}$, and define $x(t) = \Pi(y(t))$. By construction, there exists a smooth curve $\psi(t) \in \mathcal{E}_{x(t)}^u$ such that $y(t) = x(t) + \psi(t)$ for all $t$. Differentiating at $t = 0$ yields $\dot{y}(0) = \dot{x}(0) + \dot{\psi}(0)$. Since $x(t) \in \mathcal{M}$ and $\psi(t) \in \mathcal{E}_{x(t)}^u$, it follows that $\dot{x}(0) \in T_x\mathcal{M}$ and $\dot{\psi}(0) \in \mathcal{E}_x^u$. Writing $z = \dot{x}(0)$ and $w = \dot{\psi}(0)$, we obtain

$$\mathrm{Jac}(\Pi(x))(z + w) = \mathrm{D}\Pi_x(z + w) = z = P_x(z + w).$$

Since the choice of $y(t)$ was arbitrary, the claim follows. ∎

We are now ready to introduce a suitable "energy" associated with points in $\mathcal{U}_0$. Specifically, define the function

$$E(y) = \|\Pi(y) - y\|, \tag{E.12}$$

that is, the norm of the displacement between $y \in \mathcal{U}_0$ and its projection $\Pi(y)$ onto $\mathcal{M}$ along the unstable directions of (GF). By construction, this function is nonnegative and vanishes precisely on the center stable manifold:

$$E(y) \geq 0, \qquad \text{with equality if and only if } y \in \mathcal{M} \cap \mathcal{U}_0. \tag{E.13}$$

Combining (E.13) with the geometric properties established in Lemmas E.2 and E.3, we obtain that $E$ satisfies the hypotheses of Benaïm [5, Proposition 9.5]. Specialized to the present setting, this yields the following result.

**Proposition E.1** (5)**.** *There exist a compact neighborhood $\mathcal{U}_{\mathcal{Q}}$ of $\mathcal{Q}$, a constant $\beta > 0$, and a time horizon $\tau > 0$ such that the function*

$$V(x) = \int_0^{\tau} E(\Phi_{-t}(x))\, dt, \qquad x \in \mathcal{U}_{\mathcal{Q}}, \tag{E.14}$$

*satisfies the following properties:*

1. *For every $x \in \mathcal{U}_{\mathcal{Q}}$, the function $V$ admits a positively homogeneous, Lipschitz continuous right derivative $\nabla^+ V(x)$;[4] Moreover, $V$ is continuously differentiable on $\mathcal{U}_{\mathcal{Q}} \setminus \mathcal{M}$.*

2. *For all $x \in \mathcal{U}_{\mathcal{Q}}$, the directional derivative of $V$ along the gradient field satisfies*

$$\nabla^+ V(x)[\nabla f(x)] \leq -\beta V(x). \tag{E.15}$$

   *In particular, whenever $x \in \mathcal{U}_{\mathcal{Q}} \setminus \mathcal{M}$, we have*

$$\langle \nabla V(x), \nabla f(x) \rangle \leq -\beta V(x). \tag{E.16}$$

---

[4] Recall that a function $\phi$ admits a right derivative at $x$ if the limit $\nabla^+\phi(x)[v] \equiv \lim_{t \to 0^+} (\phi(x + tv) - \phi(x))/t$ exists for all $v \in \mathbb{R}^d$.

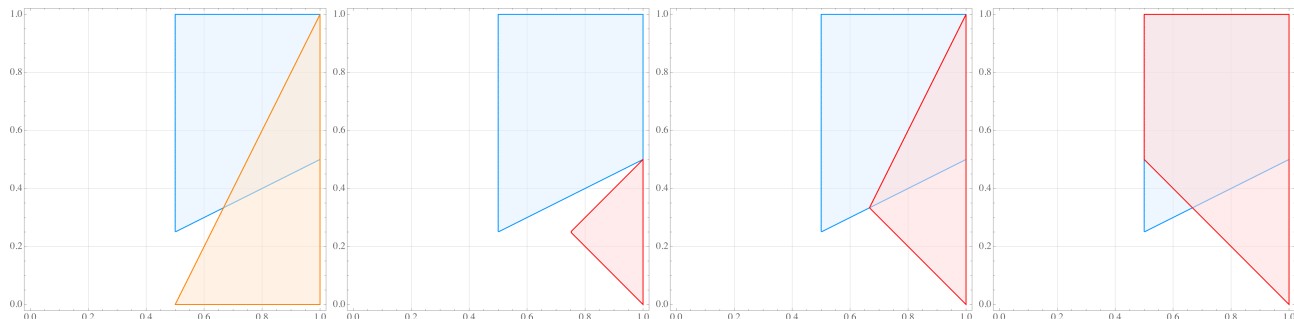

**Figure 3.** The interplay of the stepsize $\gamma_t \propto t^{-p}$ and the taming parameter $\lambda_t \propto t^{-q}$ for different values of $p$ (on the $x$-axis) and $q$ (on the $y$-axis). From the left to the right we plot the intersection of valid regions for avoidance (in blue) and almost-sure convergence (orange & red), starting from (TSGD) to (DTSGD) with $\epsilon = 1$ $\epsilon = \frac{1}{2}$ and $\epsilon \approx 0$

3. *There exists $\alpha > 0$ such that, for all $x \in \mathcal{U}_\mathcal{Q}$ and all sufficiently small $v \in \mathbb{R}^d$,*

$$V(x + v) \geq V(x) + \nabla^+ V(x)[v] - \frac{\alpha}{2}\|v\|^2. \tag{E.17}$$

4. *There exists $\beta > 0$ such that, for all $v \in \mathbb{R}^d$, the following estimates hold:*

$$\|\nabla V(x)\| \geq \beta, \qquad\qquad \text{for all } x \in \mathcal{U}_\mathcal{Q} \setminus \mathcal{M}, \tag{E.18a}$$

*and*

$$\nabla^+ V(x)[v] \geq \beta \|\mathbb{I}_x(v) - v\|, \qquad \text{for all } x \in \mathcal{U}_\mathcal{Q} \cap \mathcal{M}. \tag{E.18b}$$

Since Proposition E.1 is a direct consequence of Benaïm [5, Proposition 9.5], we omit its proof. Instead, we highlight two immediate implications that will be central to our analysis.

1. By (E.16), the energy $V(x(t))$ associated with any solution orbit $x(t)$ of (GF) grows at a (locally) geometric rate unless the orbit is entirely contained in the center stable manifold $\mathcal{M}$. Consequently, stochastic approximations of (GF) that do not remain on $\mathcal{M}$ for arbitrarily long time intervals must eventually depart from a neighborhood of $\mathcal{Q}$.

2. Inequality (E.17) underpins a discrete-time analogue of this argument. When $x_t$ lies sufficiently close to $\mathcal{Q}$, the energy before and after a stochastic gradient step obeys

$$V(x_{t+1}) \geq V(x_t) + \beta \gamma_t V(x_t) - \gamma_t \chi_t - \frac{\alpha \gamma_t^2}{2}\|\hat{v}_t\|^2, \tag{E.19}$$

where $\chi_t$ denotes an additive noise term that is non-antagonistic in expectation. In particular, the sequence $V_t \equiv V(x_t)$ exhibits, on average, locally geometric growth, precluding the possibility that the iterates $x_t$ linger near $\mathcal{Q}$ for extended periods of time.

To formalize these statements, we will rely on the probabilistic estimates established in Appendix E.1. This is the focus of the next section.

**E.3. Avoidance of saddle-point manifolds.** For ease of reference, we first restate our main avoidance statement:

**Theorem E.1.** *Let $x_t$ be the sequence of iterates generated by (RM) with step-size and bias parameters of the form $\gamma_t \propto 1/t^p$ and $\|b_t\| = \mathcal{O}(1/t^q)$, with $p \in (1/2, 1]$, $q \in (0, 1]$, and $p/2 < q \leq 1$. If $\mathcal{Q}$ is a strict saddle manifold of $f$, we have $\mathbb{P}(x_t \to \mathcal{Q} \text{ as } t \to \infty) = 0$.*

**Auxiliary results.** We first recall a key probabilistic estimate due to Pemantle [65].

**Lemma E.4.** *Let $S_t$ be a nonnegative adapted process of the form $S_t = S_0 + \sum_{s=1}^{t} X_s$, and define $\alpha_t = \sum_{s=t}^{\infty} \gamma_s^2$. Assume there exist constants $a_1, a_2 > 0$, a sequence $\varepsilon_t = o(\sqrt{\alpha_t})$, and an index $N_0$ such that, for all $t \geq N_0$:*

(i) $|X_t| = o(\sqrt{\alpha_t})$;

*(ii)* $\mathbb{1}\{S_t > \varepsilon_t\}\,\mathbb{E}[X_{t+1} \mid \mathcal{F}_t] \geq 0$;

*(iii)* $\mathbb{E}[S_{t+1}^2 - S_t^2 \mid \mathcal{F}_t] \geq a_1 \gamma_t^2$;

*(iv)* $\mathbb{E}[X_{t+1}^2 \mid \mathcal{F}_t] \leq a_2 \gamma_t^2$.

*Then* $\mathbb{P}(\lim_{t \to \infty} S_t = 0) = 0$.

Following the proof strategy of Benaïm [5], the proof of Theorem E.1 will follow by verifying the conditions of this lemma, in conjunction with Lemma E.1.

*Proof of Theorem E.1.* We begin by observing that, under our choice of step-size sequence $\gamma_t$, we have

$$\lim_{t \to \infty} \frac{\gamma_t}{\sqrt{\sum_{s=t}^{\infty} \gamma_s^2}} = 0, \tag{E.20}$$

and hence $\gamma_t = o\left(\sqrt{\sum_{s=t}^{\infty} \gamma_s^2}\right)$. This fact will be invoked below when applying Lemma E.4 with $\varepsilon_t = \mathcal{O}(\gamma_t)$ and $\alpha_t = \sum_{s=t}^{\infty} \gamma_s^2$.

Now, let $N \in \mathbb{N}$ and assume that $X_N \in \mathcal{U}(\mathcal{K})$, where $\mathcal{U}(\mathcal{K})$ is the neighborhood provided by Proposition E.1. Define the (first) exit time from $\mathcal{U}(\mathcal{K})$ by

$$T := \inf\{s \geq N : X_s \notin \mathcal{U}(\mathcal{K})\}. \tag{E.21}$$

Clearly, $T$ is a stopping time with respect to the filtration $\mathcal{F}_t$. Thus, to prove Theorem E.1, it suffices to show that $\mathbb{P}(T < \infty) = 1$. Without loss of generality, we take $N = 0$.

We now introduce two adapted sequences $\{X_t : t \geq 0\}$ and $\{S_t : t \geq 0\}$ by

$$X_{t+1} = (\eta(x_{t+1}) - \eta(x_t))\,\mathbb{1}\{t \leq T\} + \gamma_t\,\mathbb{1}\{t > T\}, \tag{E.22a}$$

$$S_0 = \eta(X_0), \qquad S_t = S_0 + \sum_{s=1}^{t} X_s. \tag{E.22b}$$

In particular, $S_t \geq 0$ almost surely for every $t$. The remainder of the proof consists in verifying conditions (i)—(iv) of Lemma E.4.

**Verification of Lemma E.4 (i) and (iv).** Since $\eta$ is Lipschitz on $\mathcal{U}(\mathcal{K})$, there exists $L' > 0$ such that

$$|\eta(x_t) - \eta(x_{t+1})| \leq L' \|x_t - x_{t+1}\|$$
$$= \gamma_t \|v(x_t) + Z_t + b_t\|. \tag{E.23}$$

Now, by assumption, we have

$$\|b_t\| = \mathcal{O}(1/t^q) \tag{E.24}$$

as long as $x_t$ remains in a closed neighborhood of crit $f$ (and hence compact, since crit $f$ is bounded). Therefore, invoking Assumption 5, we conclude that $|X_{t+1}| = \mathcal{O}(\gamma_t) = o(\sqrt{\alpha_t})$, which yields Lemma E.4 (i) and (iv).

**Verification of Lemma E.4 (ii).** Let $k' = k\|v\| + K$, where $k$ is the constant in Proposition E.1(c), $\|v\| := \sup\{\|v(x)\| : x \in \mathcal{U}(\mathcal{K})\}$, and $K$ is the uniform bound of $\|Z_t\|$. Assume $t \leq T$. Then, by Proposition E.1(ii)—(vi), we obtain

$$\eta(x_{t+1}) - \eta(x_t) \geq \gamma_t \nabla^+ \eta(x_t)[v(x_t) + Z_t + b_t] - k\gamma_t^2 (\|v\| + \|Z_t\| + \|b_t\|)^2$$
$$\geq \gamma_t \beta \eta(x_t) + \gamma_t \nabla^+ \eta(x_t)[Z_t] + \gamma_t \nabla^+ \eta(x_t)[b_t] - 2k'\gamma_t^2 - 2k\gamma_t^2 \|b_t\|^2. \tag{E.25}$$

Furthermore, by our assumption on the bias, together with Assumption 5 and the (local) Lipschitz continuity of $f$, there exists $c' > 0$ such that

$$-\|b_t\| \geq -c'\gamma_t \qquad \text{a.s.} \tag{E.26}$$

Using also the Lipschitz continuity of $\eta$, we may absorb the last three terms in (E.25) into a single quadratic remainder and obtain

$$\eta(x_{t+1}) - \eta(x_t) \geq \gamma_t \beta \eta(x_t) + \gamma_t \nabla^+ \eta(x_t)[Z_t] - 2k'' \gamma_t^2, \tag{E.27}$$

for some constant $k'' > 0$.

Therefore,

$$\mathbb{1}\{t \leq T\} \, \mathbb{E}[X_{t+1} \,|\, \mathcal{F}_t] \geq \mathbb{1}\{t \leq T\} \big[ \gamma_t \beta \eta(x_t) - 2k'' \gamma_t^2 + \gamma_t \, \mathbb{E}[\nabla^+ \eta(x_t)[Z_t] \,|\, \mathcal{F}_t] \big]. \tag{E.28}$$

By Proposition E.1(b) and the assumption that the noise is conditionally zero-mean,

$$\mathbb{E}[\nabla^+ \eta(x_t)[Z_t] \,|\, \mathcal{F}_t] \geq \nabla^+ \eta(x_t)[\mathbb{E}[Z_t \,|\, \mathcal{F}_t]] = 0. \tag{E.29}$$

Combining (E.28) and (E.29) yields

$$\mathbb{1}\{t \leq T\} \, \mathbb{E}[X_{t+1} \,|\, \mathcal{F}_t] \geq \mathbb{1}\{t \leq T\} \big[ \gamma_t \beta \eta(x_t) - 2k'' \gamma_t^2 \big]. \tag{E.30}$$

If $t > T$, then $X_{t+1} = \gamma_t$ and hence trivially

$$\mathbb{1}\{t > T\} \, \mathbb{E}[X_{t+1} \,|\, \mathcal{F}_t] \geq 0. \tag{E.31}$$

Together, these inequalities show that Lemma E.4(ii) holds with $\varepsilon_t = (k''/\beta)\gamma_t$.

**Verification of Lemma E.4 (iii).**  We start from the identity

$$\mathbb{E}[S_{t+1}^2 - S_t^2 \,|\, \mathcal{F}_t] = \mathbb{E}[X_{t+1}^2 \,|\, \mathcal{F}_t] + 2S_t \, \mathbb{E}[X_{t+1} \,|\, \mathcal{F}_t]. \tag{E.32}$$

If $S_t \geq \varepsilon_t$, then the right-hand side of (E.32) is nonnegative by Lemma E.4(ii). If instead $S_t < \varepsilon_t$, then (E.30) and (E.31) imply $S_t \, \mathbb{E}[X_{t+1} \,|\, \mathcal{F}_t] \geq -\varepsilon_t k'' \gamma_t^2 = -\mathcal{O}(\gamma_t^3)$. Hence,

$$\mathbb{E}[S_{t+1}^2 - S_t^2 \,|\, \mathcal{F}_t] \geq \mathbb{E}[X_{t+1}^2 \,|\, \mathcal{F}_t] - \mathcal{O}(\gamma_t^3). \tag{E.33}$$

Thus, it remains to show that $\mathbb{E}[X_{t+1}^2 \,|\, \mathcal{F}_t] \geq b_1 \gamma_t^2$ for some $b_1 > 0$ and all sufficiently large $t$.

From (E.27) we obtain

$$\mathbb{1}\{t \leq T\} \Big[ \mathbb{E}[X_{t+1}^+ \,|\, \mathcal{F}_t] - \Big( \gamma_t \, \mathbb{E}[(\nabla^+ \eta(x_t)[Z_t])^+ \,|\, \mathcal{F}_t] - k'' \gamma_t^2 \Big) \Big] \geq 0. \tag{E.34}$$

By Proposition E.1 and Assumption 5, we have

$$\mathbb{1}\{t \leq T \wedge x_t \notin \mathcal{A}\} \Big( \mathbb{E}[(\nabla^+ \eta(x_t)[Z_t])^+ \,|\, \mathcal{F}_t] - c_1 b \Big) \geq 0. \tag{E.35}$$

If $x_t \in \mathcal{A}$, choose a unit vector $v_t \in \ker(I - \mathrm{Jac}\, P(x_t))^\perp$. By construction, $\langle Z_t, v_t \rangle = \langle Z_t - \mathrm{Jac}\, P(x_t) Z_t, v_t \rangle$. Let $\mathcal{H} = \{t \leq T\} \cap \{x_t \in \mathcal{A}\}$. Then, by Proposition E.1, the Cauchy–Schwarz inequality, and Assumption 5, we obtain

$$\begin{aligned}
\mathbb{1}\{\mathcal{H}\} \, \mathbb{E}[(\nabla^+ \eta(x_t)[Z_t])^+ \,|\, \mathcal{F}_t] &\geq c_1 \, \mathbb{1}\{\mathcal{H}\} \, \mathbb{E}[\|Z_t - \mathrm{Jac}\, P(x_t) Z_t\| \,|\, \mathcal{F}_t] \\
&\geq c_1 \, \mathbb{1}\{\mathcal{H}\} \, \mathbb{E}[\langle Z_t - \mathrm{Jac}\, P(x_t) Z_t, v_t \rangle^+ \,|\, \mathcal{F}_t] \\
&= c_1 \, \mathbb{1}\{\mathcal{H}\} \, \mathbb{E}[\langle Z_t, v_t \rangle^+ \,|\, \mathcal{F}_t] \\
&\geq c_1 b \, \mathbb{1}\{\mathcal{H}\}. 
\end{aligned} \tag{E.36}$$

Combining (E.34)–(E.36) (and using also (E.31)) gives

$$\mathbb{E}[X_{t+1}^+ \,|\, \mathcal{F}_t] \geq c_1 b \, \gamma_t - k'' \gamma_t^2. \tag{E.37}$$

Finally, Jensen's inequality yields $\mathbb{E}[X_{t+1}^2 \,|\, \mathcal{F}_t] \geq \mathbb{E}[X_{t+1}^+ \,|\, \mathcal{F}_t]^2$, so $\mathbb{E}[X_{t+1}^2 \,|\, \mathcal{F}_t] \geq b_1 \gamma_t^2$ for some $b_1 > 0$ and all large enough $t$. Together with (E.33), this verifies Lemma E.4(iii).

**Closing the argument.** We have now verified Lemma E.4(i)—(iv), and therefore

$$\mathbb{P}(\lim_{t \to \infty} S_t = 0) = 0. \tag{E.38}$$

We use (E.38) to conclude that $T < \infty$ almost surely.

Indeed, suppose $T = \infty$. Then $X_{t+1} = \eta(x_{t+1}) - \eta(x_t)$ and $S_t = \eta(x_t)$ by (E.22a)–(E.22b), and the iterates remain in $\mathcal{U}(\mathcal{K})$ by definition of $T$. By assumption, the limit set $L(\{x_t\})$ is a nonempty compact invariant subset of $\mathcal{U}(\mathcal{K})$; in particular, $\Phi_t(x') \in \mathcal{U}(\mathcal{K})$ for all $x' \in L(\{x_t\})$ and all $t \in \mathbb{R}$. But then Proposition E.1(iv) implies that $\eta(\Phi_t(x')) \geq e^{\beta t} \eta(x')$ for all $t > 0$, forcing $\eta(x') = 0$. Hence $L(\{x_t\}) \subseteq \mathcal{A}$, and therefore $S_t = \eta(x_t) \to 0$. By (E.38), this event has probability 0, showing that $T$ is finite almost surely. ∎

# F    Numerics

In this Section, we provide a complementary character to the theoretical findings of this work, by illustrating the behavior of (TSGD) and comparing it with some similar benchmark methods on two simple numerical experiments.

**F.1. Phase retrieval.** In the first experiment, we aim to solve a phase retrieval problem, which consists of recovering a source vector $x_* \in \mathbb{R}^d$, from a finite number $N$ of quadratic measurements, cf. [15]. In this context, we consider the minimization problem (Opt):

$$f(x) = \frac{1}{N} \sum_{i=1}^{N} f_i(x), \qquad f_i(x) = \frac{1}{4} \left( (w_i^\mathsf{T} x)^2 - y_i \right)^2, \tag{F.1}$$

where $(w_i)_{i \leq N} \in \mathbb{R}^d$ are the sampling (Gaussian) vectors and $y_i = (w_i^\mathsf{T} x_*)^2$ the observed measurements from an unknown signal $x_* \in \mathbb{R}^d$ drawn from $\mathcal{N}(0, I_d)$, with $d = 10$ and $N = 200$. The function defined in (F.1) is smooth, non convex, with gradient growth of order 3 (i.e. $r(x) = \|x\|^3$).

In Figure 4 we illustrate and compare the performance of (TSGD), with three different choices for the scale function $r(x)$ ($r(x) = \|x\|^3$, $r(x) = \|x\|^3$ and $r(x) = \exp(\|x\|)$), with vanilla (SGD) and clipped stochastic (CGD) with clipping level equal to 4. For each method, the stochastic oracle at each iteration is computed as $g_t = \nabla f_M(x_t) + \xi_t$, where $\nabla f_M(x_t) = \frac{1}{|M|} \sum_{i \in M} \nabla f_i(x_t)$ is a mini-batch estimator of $\nabla f(x_t)$, with $|M| = 48$ and $\xi_t \sim \mathcal{N}(0, \sigma^2 I_d)$ is centered Gaussian perturbation, with variance $\sigma$ measuring the level of extra noise injected to $\nabla f_M(x_t)$. In addition the step-size $\gamma_t$ and the taming parameter $\lambda_t$ (for (TSGD)), are set as $\gamma_t = t^p$ and $\lambda_t = t^{-q}$, with $p \sim 0.6$ and $q \sim 0.1$, for all schemes. The performance is measured in terms of success rate for achieving $\|f(x_t)\| < \varepsilon$, with $\varepsilon = 0.001$ over 80 runs, across different i) maximum number of total iterations and ii) level of extra noise $\sigma$ added to the stochastic approximation of $\nabla f(x_t)$.

From Figure 4, we observe that all three instances of (TSGD) are outperforming vanilla (SGD), rendering the taming mechanism beneficial for tackling such an optimization objective function with superlinear growing gradients. Furthermore, the "tight" version of (TSGD), corresponding to the exact gradient growth of the objective function in (F.1) (i.e. $r(x) = \|x\|^3$) performs better over the more pessimistic choices corresponding to $r(x) = \|x\|^5$ and $r(x) = \exp(\|x\|)$, and over clipped stochastic (CGD), both in terms of convergence speed (Fig. 4 left), as also in terms of robustness to the gradient noise (Fig. 4 right).

**F.2. Diagonal deep Neural Network.** In this example, we test numerically the behavior of (TSGD) and compare it with (SGD) and clipped-stochastic (CGD), in terms of stability to the step-size choice (initialization $\gamma_0$), on the training of a simple synthetic deep diagonal neural network by using a regularized ERM problem.

We consider a finite-sum nonconvex optimization problem (Opt), arising from the training of a deep diagonal network with regularization, where

$$f(x) = \frac{1}{N} \sum_{i=1}^{N} f_i(x), \qquad f_i(x) = \frac{1}{2} (h(w_i; x) - y_i)^2 + \frac{\mu}{2} \|x\|_F^2, \tag{F.2}$$

with $\mu > 0$ and the prediction model is given by

$$h(w; x) = \sum_{k=1}^{d} \left( \prod_{\ell=1}^{L} x_{\ell k} \right) w_k \tag{F.3}$$

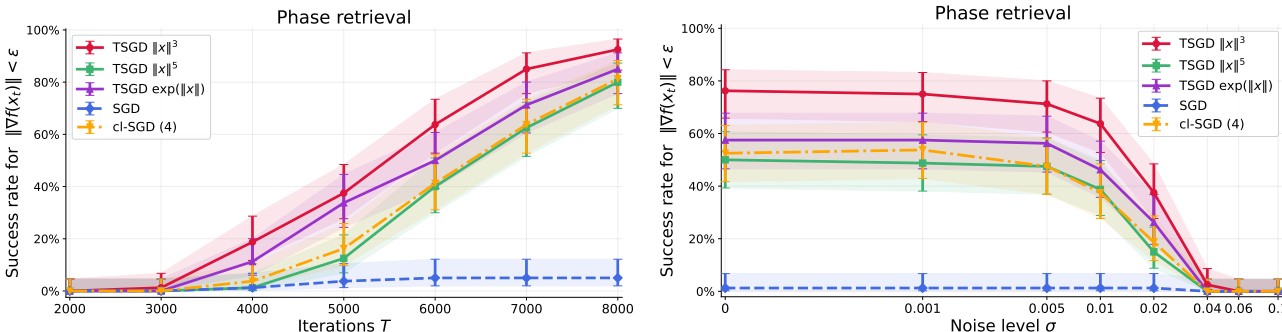

**Figure 4.** Comparison of (TSGD) with $r(x) = \|x\|^3$ (red), $r(x) = \|x\|^5$ (green) and $r(x) = \exp(\|x\|)$ (purple), with vanilla (SGD) (blue) and clipped stochastic (CGD) (orange). Left: Performance in terms of success rate across different maximum number of iterations $T$ Right: Performance in terms of success rate (number for $\|\nabla f(x_t)\| \leq 0.001$), across different extra noise levels on the stochastic oracle measured by the parameter $\sigma$ (from light $\sigma = 0.001$ to heavy $\sigma = 0.1$), over 80 runs (initializations).

The components $f_i$ are formed by generating a synthetic dataset $\{(w_i, y_i)\}_{i=1}^N$ ($w_i$ are sampled uniformly on the unit ball) and a source-target $x_* \in \mathbb{R}^{L \times d}$ and setting $y_i = h(w_i, x_*)$, with $d = 5$, $N = 128$ and $L = 4$. The resulting function $f$ in (F.2) is smooth non convex with gradient growth of order $2L - 1$ (i.e. $r(x) = \|x\|^7$). In this experiment, we test the stability performance of (TSGD), in terms of robustness to the initialization of the step-size $\gamma_0$ for three different noise levels on the stochastic estimator of the gradient, cf. [4, 77]. In particular, at every iteration we use the stochastic oracle $g_t = \nabla f_{i_t}(x_t) + \xi_t$, where $i_t \sim \mathcal{U}([1, \ldots, N])$ ($i_t$ is sampled uniformly in $[1, \ldots, N]$) and $\xi_t \sim \mathcal{N}(0, \sigma^2 \mathrm{Id})$, with $\sigma$ measuring the additional level of noise. In this setting, we compare (TSGD) with two instances of clipped (SGD) (with two different clipping levels) (CGD) and vanilla (SGD), with common stepsize for each method $\gamma_t = \gamma_0 t^{-p}$ ($p \sim 0.6$), taming parameter $\lambda_t = \lambda_0 t^{-q}$ ($q \sim 0.2$) and scale function chosen as $r(x) = \|x\|^{2L-1}$, where $L = 4$ is the depth of the network.

In Figure 5 we test the stability of these methods in terms of number of iterations to achieve a given accuracy ($\|\nabla f(x_t)\| \leq \varepsilon$, $\varepsilon \sim 0.005$)[5], as a function of the initialization of the stepsize $\gamma_0$[6], for three additional noise levels $\sigma = 0$ (no noise), $\sigma = 0.04$ (lighter noise) and $\sigma = 0.08$ (heavier noise). The (color)boxes represent the performance each tested method over 50 runs and encapsulate the interquartile range (25%-75% percentile) and the whiskers represent the min-max range. Medians are depicted with colored circles and are connected with thick lines across values of $\gamma_0$ (each color corresponds to a different method).

From Figure 5, we can observe that in all the three cases of the noise level, for small initialization of the stepsize (e.g. $\gamma_0 \in (2.5, 14)$ the (TSGD) scheme, performs at least as well as (if not better) the best instance of the clipped (SGD) and better than the vanilla (SGD) method, which fails to achieve the desired accuracy in most of the cases. For larger initializations, e.g. $\gamma_0 \in (14, 25)$, (TSGD) seems to be essentially more robust in terms of stepsize sensitivity than the rest of the methods, in all the three cases of the noise level.

---

[5]In the experiment the full gradient $\nabla f(x_t)$ is evaluated every 50 iterations.
[6]A maximum cap of 14000 iteration is also imposed in the case of not reaching the desired accuracy.

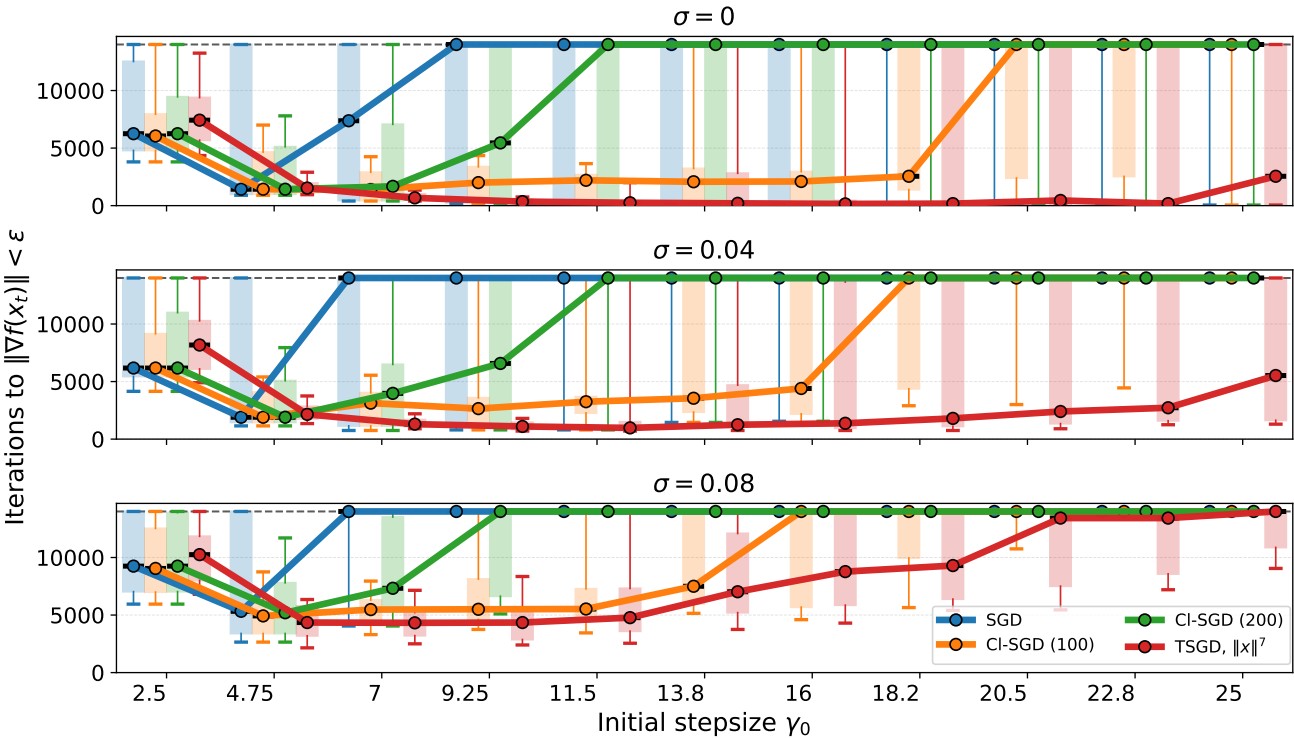

**Figure 5.** Number of iterations needed in order to achieve an accuracy of at least 0.005 (with a maximum cap at 14000 iterations), in terms of $\|\nabla f(x_t)\|$, for different initialization of the stepsize $\gamma_0$, when solving the regularized ERM problem with diagonal network (F.2). Here we compare (TSGD) (red) with clipped stochastic CGD with two clipping levels (orange=100 & green=200) and vanilla (SGD) (blue) for three different level of additional noise added to the stochastic oracle via the parameter $\sigma$, with $\sigma = 0$ (first figure-no noise), $\sigma = 0.04$ (2nd figure-lighter noise) and $\sigma = 0.08$ (3rd figure-heavier noise). Each box summarizes 50 independent runs; boxes indicate interquartile range, whiskers indicate the min-max range, and colored lines connect the medians. Runs that fail to reach the tolerance are assigned the cap value $T = 14000$.

