# OpenReview forum: "Taming Stochastic Gradient Descent: Almost Sure Convergence and Saddle-Point Avoidance under $(L_{0},L_{1})$-Smoothness"
_ICML.cc/2026/Conference — ICML 2026 regular_

### Official Review · Reviewer_BLHz · 2026-03-10

**Soundness:** 3
**Presentation:** 3
**Significance:** 3
**Originality:** 3
**Overall Recommendation:** 5
**Confidence:** 3

**Summary:**

The paper studies tamed SGD for nonconvex optimization under $(L_0, L_1)$-smoothness, which is a meaningful relaxation of standard global smoothness. It proposes two methods: Tamed SGD, which gives almost-sure convergence under stronger noise assumptions, and Dissipative Taming SGD, which relaxes part of that at the price of a more involved taming scheme. The paper also proves a strict-saddle avoidance result and derives polynomial schedule ranges for the stepsize and taming parameters. Overall, the paper aims to push SGD theory beyond the classical smooth setting while keeping asymptotic guarantees.

**Compliance With Llm Reviewing Policy:**

Affirmed.

**Final Justification:**

The authors addressed my concerns in the rebuttal, and it further strengthened my positive assessment of this work. I therefore raised my score from Weak Accept to Accept accordingly.

**Key Questions For Authors:**

1. In Appendix E "Verification of Lemma E.4": the proof seems to use a bound on $\|b_t\|$ that is stronger than the theorem statement assumes. As written, equation (E.26) appears to treat the bias as if it were of order gamma_t, which would correspond to $q \geq p$, while the theorem is stated under the weaker condition $q > p/2$. If I'm correct on the fact that this is an error, I still believe the argument can likely be repaired by keeping a mixed remainder of order $\gamma_t\lambda_t$ instead of absorbing it directly into $\gamma_t^2$. Still, this point should be corrected carefully, because if $q \geq $ p were really needed, then some of the polynomial schedule regions stated in the paper would become empty.

2. Which of the stronger assumptions do the authors believe are genuinely necessary, and which are mostly proof artifacts? In particular, how essential are gradient coercivity, the ABC condition, and uniform directional excitation?

3. Could the authors give clearer and more robust guidance for choosing the taming scheme in practice, especially for the dissipative version?

**Limitations:**

Yes

**Strengths And Weaknesses:**

**Strengths:**
The problem is relevant. Weakening assumptions for SGD is a useful direction if one wants theory to get closer to practice. The paper is also very easy to follow: the motivation is clear, everything is well detailed and the notations are standard. I also think the use of $(L_0, L_1)$-smoothness is well motivated: it is a real relaxation of global smoothness, that covers way more practical cases. I also find tying taming and clipping is enlighting and further justifies gradien these schemes.

**Weaknesses:**
The assumptions are the weakest part of the presentation. They are spread across the paper, and it is not always clear which ones are standard and which ones are truly strong. Some are usual, but others are restrictive, especially gradient coercivity, the ABC condition, and uniform directional excitation. For example, many natural objectives are not gradient-coercive unless one adds explicit L2 regularization. The practical design of the taming scheme is also not very clear from the writing, especially for the dissipative variant. Finally, the experimental section is too weak to support strong practical conclusions. There is only one synthetic experiment, and the variance bands are so wide that they are largely inconclusive

In general, I think that this paper is a good contribution. However, the paper would benefitt from a clearer presentation of the assumptions, ideally grouped in one place and ranked by how standard or restrictive they are. It would also help to give more direct practical guidance for choosing the taming scheme, especially for the dissipative method. On the empirical side, I think that further experiments could enhance the understanding of taming and of the assumptions, ideally with examples both within and outside the stated assumptions. That would make it much easier to understand what is really gained by the theory and how one should use the taming methods in practice.

---

> ### Author Rebuttal · Authors · 2026-03-31
>
> Dear reviewer,
>
> Thank you for your time and positive evaluation! We reply to your remarks and questions below:
>
> > The assumptions are spread across the paper, and it is not always clear which ones are standard and which ones are truly strong [...] How essential are gradient coercivity, the ABC condition, and uniform directional excitation?
>
> In terms of presentation, we opted to introduce each assumption as close as possible to the point where it was first used, to avoid forcing the reader to go back and forth to check each of them individually. Our grouping and choice of assumptions was also driven by a similar principle: we preferred stronger assumptions to have a simpler presentation that would be more accessible to non-experts. We understand your point however, and are happy to regroup our assumptions as suggested.
>
> As for the strength and necessity of each assumption we can provide the following additional pointers:
> - Gradient coercivity is only needed for Thm 2, it is **not needed** for Thms 1 & 3. [This can be seen from the proofs]
> - The ABC conjecture can also be relaxed significantly (especially for Thm 1), using e.g. any of the variants discussed in [2]. At any rate, this is not restrictive for the type of applications we have in mind (which are all of finite-sum form, so the ABC condition is satisfied, cf. Section 2.3 and the relevant discussions afterwards).
> - The uniform excitation is more difficult to lift. The version we employ is the classical assumption in the avoidance literature, and dates back to Pemantle, Duflo and Benaïm & Hirsch in the 90's. By more recent work of P. Tarrès et al, it can be relaxed slightly by assuming uniform excitation along the unstable manifold of a saddle point, but even this relaxation is highly challenging (and is believed to be the minimal assumption guaranteeing a.s. avoidance). All in all, this is a very interesting—and difficult!—mathematical question; we note however that, in practice, this assumption is not restrictive, and it is typically satisfied (at worst, by injecting a small amount of Gaussian noise in the method).
>
> Please let us know if any of the above is not clear.
>
> > There is only one synthetic experiment, and the variance bands are so wide that they are largely inconclusive.
>
> To clarify, the color bands represent the best/worst cases, so the overlap is the effect of the outliers; the actual variance bands are considerably narrower and do not exhibit any statistically significant overlap. To clarify, we will switch to boxplots at the first possible revision opportunity.
>
> Beyond this technical point, the simulations are not intended as an extensive numerical validation of all the different schemes, but simply as an illustration with a complementary character to our main results, providing a rough idea of how the various methods work in a simply, synthetic setting.
>
> > In Appendix E "Verification of Lemma E.4": the proof seems to use a bound on $b_t$ that is stronger than the theorem statement assumes. As written, equation (E.26) appears to treat the bias as if it were of order $\gamma_t$.
>
> You're absolutely right, thanks for pointing out. There was a series of typos: letting $\beta_t=\|b_t\|$, the $\gamma_t$ in (E.26) should be $\beta_t$, and the $\gamma_t^2$ in (E.27) and onwards should read $\gamma_t^2 + \beta_t^2$. You are right that this needs care: $\epsilon_t$ should be chosen proportionally to $\gamma_t + \beta_t$, and it must satisfy the condition $\epsilon_t = o(\sqrt{\sum_{s\geq t} \gamma_s^2})$. Fortunately, this last bound is quite lax, and this is where the parameter range reported in Thm E.1 is coming from.
>
> Thanks for spotting this, and apologies for any confusion—the author responsible has been tarred and feathered.
>
> > Could the authors give clearer and more robust guidance for choosing the taming scheme in practice, especially for the dissipative version?
>
> Thanks for raising this point. In a nutshell:
> - For $p$ and $q$, we recommend taking $p$ as large as permitted by (22). For $\epsilon$, we recommend taking it as small as possible, in order to allow the fastest possible step-sizes (cf. Fig. 2, App E).
> - For the scale function $r(x)$, please see our reply to [Reviewer 3VPk](https://openreview.net/forum?id=v7i8l6FXh9&noteId=Oo9NlL267a) above.
> - For the choice of $a$ in DTSGD, any value would do, but we would recommend setting it small. If $f$ is known to satisfy a condition of the form $\langle \nabla f(x),x \rangle \geq \alpha \|x\|^2 - \beta$ for some known $\alpha>0$, we recommend $a = \alpha$ (as this would lead to the most contractive algorithm).
> We understand these guidelines are important from a practical standpoint, and we will amend our presentation accordingly.
>
> ---
> Thanks again for your time and positive evaluation. We will of course be happy to include the points discussed above at the first possible revision opportunity—please let us know if you have any further questions in the meantime.
>
> Kind regards,
> The authors

---

> > ### Author Rebuttal · Reviewer_BLHz · 2026-04-01
> >
> > I thank the authors for their reply, my concerned have been adressed.
> > I think this is a valuable contribution and increase my score accordingly.

---

> > > ### Author Response · Authors · 2026-04-02
> > >
> > > Dear reviewer BLHz,
> > >
> > > Thank you for your reply, your kind words, and your increased support, we are very happy that your concerns were addressed.
> > >
> > > Kind regards,
> > >
> > > The authors

---

### Official Review · Reviewer_rYuV · 2026-03-12

**Soundness:** 1
**Presentation:** 2
**Significance:** 3
**Originality:** 3
**Overall Recommendation:** 3
**Confidence:** 4

**Summary:**

In this paper, the authors present the TSGD method, which converges almost surely under the assumption of pointwise sub-Gaussian noise with the ABC assumption (an affine parameter $\sigma(x)$) in the case of generalized smoothness. The authors also provide a modification of this method (DTSGD) and additionally analyze the avoidance of saddle points.

**Compliance With Llm Reviewing Policy:**

Affirmed.

**Final Justification:**

At the current moment, I have only one concern -- generalizability of the proposed approach related to $r(x)$. At the same time, the paper is interesting, and almost all concerns were adequately addressed. Therefore, I increased my final evaluation to "weak reject".

**Key Questions For Authors:**

1) Do you think it is possible to modify $r(x)$ so that it can be used in practical sphere (i.e., without deterministic evaluations of the function/gradient and without knowing $f^*$)?

**Limitations:**

Yes

**Strengths And Weaknesses:**

## Strenghts

1) **Interesting results.** The obtained results are interesting and useful, and they can also be further developed in future work.

## Weaknesses

Unfortunately, while reading the paper, some shortcomings stand out.

>**Presentation.**

From the perspective of presentation and the clearness of some parts of the proof, there are issues. In particular, one can point out next shortcomings:

-- "thanks"; probably, it should be "due to";

-- Definition of $(L_0, L_1)$, page 2 -- it is for *twice* differentiable functions;

-- "Broken" citation; on page 12 in the section "Further related work" in the part about taming scheme; Additionally, I recommend to look at these papers [1, 2]: both of them provide high-probability analysis for Clipped-SGD for the generalized smooth case under different assumptions on the stochasticity (sub-Gaussian and heavy-tailed noise, respectively) - it could supplement the literature review;

-- Definition of Normalized Gradient Descent: it is not NGD in the primal form ($\beta$ must be equal to $0$). Moreover, both equations (10 and NGD) describe very similar methodologies;

-- Proof of Theorem 1, C.6 -- brackets are needed in the third line in the exponent and in the application of convexity of $|| \cdot ||^2$;

-- Proof of Theorem 1, C.13 -- Most likely, the conditional expectation is missing (I believe it should be $\mathbb{E}_t$ instead of $\mathbb{E}$; otherwise, the unconditional expectation would be upper-bounded by a stochastic random variable $r(x_s)$);

-- Lemma D.2 -- it should be $||\Delta_t||^2$ instead of $||\Delta_t||$ in the formulation of the lemma. What is more, in the start of the proof, $2$ is forgotten as a coefficient of the scalar product;

Based on the above, I recommend that the authors double-check the paper for such issues. I would like to emphasize that I consider these comments to be minor - they are merely suggestions to improve the presentation of the paper and the readability of the proofs.

>**Unclear parts of the proof.**

During my review of the proof, I came across several points that remain unclear, namely:

1) Theorem 1, C.7. It is not clear for me why it is guaranteed that $L_0 + L_1||\nabla f(x_t)|| \leq \frac{C_1}{\lambda_t}(1 + \lambda_t r(x_t))$. This fact should be proven; otherwise, next parts of the proof seem incorrect.
2) Theorem 1, C.13. It is not obvious why expression C.13 implies almost-sure boundedness of quantity $v_{\infty}$. This claim is either incorrect or requires additional justification.

>**Impracticality.**

Despite the interesting idea considered by the authors, the choice in $r(x)$ remains very confusing. At the very least, the following is completely unclear: why use a stochastic gradient if computing $r(x)$ requires knowledge of the full gradient? If one has access to a full-gradient oracle, then it seems unnecessary to compute the stochastic gradient to determine the update direction in TSGD/DTSGD. Moreover, computing $r(x)$ requires knowledge of $f^*$ - essentially the same issue as with the Polyak-Shor stepsize. Therefore, using $r(x)$ as in expression (12) appears impractical. I understand that the authors provide a fairly clear example with feed-forward neural networks, but it remains unclear how to choose $r(x)$ in the general case.

---

### References

[1] Gaash et al., Convergence of Clipped SGD on Convex $(L_0,L_1)$-Smooth Functions. 2025

[2] Chezhegov et al., Convergence of Clipped-SGD for Convex $(L_0,L_1)$-Smooth Optimization with Heavy-Tailed Noise, 2025

---

> ### Author Rebuttal · Authors · 2026-03-31
>
> Dear reviewer,
>
> Thank you for your time. We reply to your remarks and questions below:
>
> > Despite the interesting idea considered by the authors, the choice in $r(x)$ remains very confusing: why use a stochastic gradient if computing $r(x)$ requires knowledge of the full gradient? [...] Moreover, computing $r(x)$ requires knowledge of $f^\ast$, essentially the same issue as with the Polyak-Shor stepsize. [...] Do you think it is possible to modify $r(x)$ so that it can be used in practical sphere (i.e., without deterministic evaluations of the function/gradient and without knowing $f^\ast$)?
>
> We are concerned there may be a misunderstanding here: **tuning $r(x)$ does not require knowledge of full gradients or $\min f$, but only an upper bound on the growth of $\|\nabla f(x)\|$ as $\|x\|\to\infty$.** This is why the tuning in (12) and (17) has been stated in asymptotic $\Omega(.)$ notation: exact values are irrelevant, only a bound on the growth of $\nabla f(x)$  is required.
>
> Regarding the practicality of the proposed taming scheme, here are some concrete examples:
> - **In deep learning** (cf. the discussion in L294, right before Theorem 1): in an $L$-layer feed-forward neural network with any of the standard activation functions (ReLU, softplus, sigmoid,...), gradients grow as $\|x\|^{2L-1}$, so the right tuning is $$r(x) = 1 + \|x\|^{2L-1}$$
> - **In matrix factorization problems** (including matrix sensing, matrix completion, and robust PCA models), gradients grow cubically in the problem's control variable (a matrix), so the tuning here is $$r(x)=1 + \|x\|^3$$
> - **In phase retrieval problems** (with a least square objective), gradients again exhibit cubic growth, so the right tuning is $$r(x) = 1+\|x\|^3$$
> - **In tensor PCA / rank-one approximation problems**, the objective is an $m$-th or $(2m)$-th degree polynomial (for homogeneous and least-square fitting respectively), so the right tuning for $r$ is $$r(x) = 1 + \|x\|^{\{m,2m\}-1}$$
> - **Black-box:** finally, if the optimizer has **no information whatsoever about $f$**, the $(L_0,L_1)$-smoothness condition suggests the tuning $$r(x)=\exp(x^{1+\epsilon})$$with $\epsilon>0$ chosen arbitrarily ($\epsilon=1/2$ is a reasonable, default choice). This case is not very relevant from a practical standpoint, but it is still important as a theoretical worst-case instance (which is why the setting of $r(x)$ is so conservative in this case).
> To sum up:
> - **(D)TSGD does not require knowledge of full gradients at any point.**
> - **The tuning of $r(x)$ can be achieved in a very practical manner, even in the black-box case.**
>
> > Theorem 1, C.7. It is not clear for me why it is guaranteed that $L_0 + L_1 \|\nabla f(x_t)\| \leq C_1 (1+\lambda_t r(x_t))/\lambda_t$.
>
> This is a direct consequence of the boundedness of $\lambda_t$ and (12), i.e. the fact that $r(x)=\Omega(\|\nabla f(x_{t})\|)$). Thus, with $\|\nabla f(x)\| \leq K_{0}r(x)$ and $\lambda_{t}\leq K_{1}$ for some $K_{0},K_{1}>0$, we have
> $$
> L_{0}+L_{1}\|\nabla f(x_{t})\| \leq \frac{L_{0}K_{1}}{\lambda_{t}} + \frac{L_{1}K_{0}\lambda_{t}}{\lambda_{t}}r(x_{t}) \leq \frac{\max\{L_{0}K_{1},L_{1}K_{0}\}}{\lambda_{t}}(1+\lambda_{t}r(x_{t}))
> $$
> which is the claimed inequality with $C_{1}=\max\{L_{0}K_{1},L_{1}K_{0}\}$.
>
> > Theorem 1, C.13. It is not obvious why C.13 implies almost-sure boundedness of $v_\infty$.
>
> This is an immediate consequence of the martingale convergence theorem, a textbook result in probability theory (see e.g., Durrett's book, *Probability: Theory and Examples,* Thm 4.2.11). The theorem states that a martingale that is bounded in $L^p$  for some $p\geq 1$ converges almost surely to some a.s. finite random variable. We will add a precise reference—thanks for bringing this up.
>
> > Definition of Normalized Gradient Descent: it is not NGD in the primal form (β must be equal to 0). Moreover, both equations (10 and NGD) describe very similar methodologies.
>
> As we discuss in p. 5 (L272-274), there is a discrepancy in the literature regarding the term "normalized": we follow here the convention of Zhang et al. [82], while also mentioning that (NGD) is also encountered with $\beta= 0$, see e.g., [15,77]. While we agree that the terminology "normalized" fits more naturally to the case $\beta = 0$, we chose the convention that seems to have appeared first in the literature. We do not otherwise take a position in this debate.
>
> > I recommend to look at [1, 2].
>
> Thanks for the pointers. Both papers concern the convex case, but we will be happy to discuss them.
>
> > [Minor comments and typos]
>
> We will take care of those, thanks for pointing them out.
>
> ---
> Thank you again for your time. We will of course be happy to incorporate the above points at the first revision opportunity. In the meantime, we hope and trust that our replies have resolved your concerns and we look forward to any further questions you may have.
>
> Kind regards,
>
> The authors

---

> > ### Author Rebuttal · Reviewer_rYuV · 2026-04-03
> >
> > Thanks to the authors for the thorough response. Almost all of the concerns that arose while reading the paper can be considered resolved. However, I have only one remaining question.
> >
> > It is claimed that we only need to have $\Omega(||\nabla f(x)||)$ as $||x|| \to \infty$. However, there may be a mismatch between the practical results and the theoretical guarantees. Based on formula (12) form the paper, condition $\Omega(\sqrt{f(x) - f^\star})$ also appears to be necessary. Unfortunately, a lower bound on the gradient norm alone does not seem sufficient here. As a result, the following question arises: if the practical examples you provide satisfy guarantees of type $\Omega(||\nabla f(x)||)$, do these guarantees also extend to $\Omega(\sqrt{f(x) - f^\star})$? If so, I will be ready to reconsider my final score; otherwise, the theory and the practical examples do not appear to be consistent with each other.
> >
> > # Update
> >
> > Thank you for your response. It seems to me that there is currently an issue with the generalizability of this approach - even if it works in illustrative examples, the statement may be false in full generality (this concerns expression $r(x)$). Nevertheless, the paper itself is interesting, and so is the proof technique.
> >
> > Based on it, I am ready to raise my score to 3.
> >
> > Best regards,
> >
> > Reviewer

---

> > > ### Author Response · Authors · 2026-04-05
> > >
> > > Dear Reviewer rYuV,
> > >
> > > Thank you for your time and follow-up.
> > >
> > > > However, there may be a mismatch between the practical results and the theoretical guarantees. [...] The following question arises: if the practical examples you provide satisfy guarantees of type $\Omega(|\nabla f(x)|)$, do these guarantees also extend to $\Omega(\sqrt{f(x) - f^\ast})$? If so, I will be ready to reconsider my final score.
> > >
> > > Yes indeed, they do. Please allow us to clarify:
> > > - **In the deep learning example:** the loss function grows polynomially as $f(x)=\mathcal{O}(|x|^{2L})$ where $L$ is the number of layers in the network. As a result, $\sqrt{f(x) - f^\ast}$ grows as $\mathcal{O}(|x|^L)$ and $|\nabla f(x)|$ grows as $\mathcal{O}(|x|^{2L-1})$, so the indicated choice $r(x) = 1 + |x|^{2L-1}$ bounds both $|\nabla f(x)|$ and $\sqrt{f(x) - f^\ast}$.
> > > - **In the matrix factorization example:** here, the problem's loss function is quartic, i.e., it grows as $f(x) = \mathcal{O}(|x|^4)$. Accordingly, $\sqrt{f(x) - f^\ast}$ grows as $\mathcal{O}(|x|^2)$ and $|\nabla f(x)|$ grows as $\mathcal{O}(|x|^{3})$, so the indicated choice $r(x) = 1 + |x|^{3}$ bounds both terms.
> > > - **In the phase retrieval example:** the problem's loss function is, again, quartic, i.e., $f(x) = \mathcal{O}(|x|^4)$. Consequently, $\sqrt{f(x) - f^\ast}$ grows as $\mathcal{O}(|x|^2)$ and $|\nabla f(x)|$ grows as $\mathcal{O}(|x|^{3})$, so the indicated choice $r(x) = 1 + |x|^{3}$ bounds both terms.
> > > - **In the tensor approximation example, least squares fitting:** the problem's loss function grows as a $(2m)$-th degree polynomial, where $m\geq2$ is the order of the tensor being approximated. As a result, $\sqrt{f(x) - f^\ast}$ grows here as $\mathcal{O}(|x|^m)$ and $|\nabla f(x)|$ grows as $\mathcal{O}(|x|^{2m-1})$, so the indicated choice $r(x) = 1 + |x|^{2m-1}$ bounds both terms.
> > > - **In the tensor approximation example, homogeneous fitting:** in this case, the problem's loss function grows as an $m$-th degree polynomial, so $\sqrt{f(x) - f^\ast}$ grows as $\mathcal{O}(|x|^{m/2})$ and $|\nabla f(x)|$ grows as $\mathcal{O}(|x|^{m-1})$. As a result, the indicated choice $r(x) = 1 + |x|^{m-1}$ bounds both terms. [We realize now that the OpenReview markdown did not compile braces correctly, which may have led to confusion in this example.]
> > >
> > > So, to conclude: **yes, in all the practical examples that we provide, the indicated choice for $r(x)$ also covers the $\Omega(\sqrt{f(x) - f^\ast})$ requirement.**
> > >
> > > To provide some more context, in all of the practical examples that we provided, the gradient norm is the dominant term—that is, $\sqrt{f(x) - f^\ast} = \mathcal{O}(|\nabla f(x)|)$, and this was the sense in which we stated in our reply that "only an upper bound on the growth of $|\nabla f(x)|$ is required". In the general $(L_0,L_1)$ case, the agnostic, black-box choice $r(x)=\exp(x^{1+\epsilon})$ still applies and bounds *both* $|\nabla f|$ and $\sqrt{f}$ (though, as we mentioned above, this tuning is primarily of theoretical interest and overly conservative in practice).
> > >
> > > ---
> > > We hope this serves to provide a clearer picture. While OpenReview does not allow further back-and-forth at this stage, we would be happy to provide additional clarifications via the AC if needed.
> > >
> > > Thank you again for your time. Kind regards,
> > >
> > > The authors

---

### Official Review · Reviewer_Napj · 2026-03-13

**Soundness:** 3
**Presentation:** 3
**Significance:** 3
**Originality:** 3
**Overall Recommendation:** 5
**Confidence:** 3

**Summary:**

This paper studies the almost sure convergence of stochastic gradient descent (SGD) trajectories
under a taming scheme when the objective function is locally, but not globally, Lipschitz smooth.
The authors analyze the behavior of the tamed SGD dynamics and establish convergence results
under these relaxed smoothness conditions. In addition, the paper shows that the proposed
taming mechanism prevents convergence to certain non-minimizing saddle points. These results
aim to extend the theoretical understanding of SGD convergence beyond the classical globally
Lipschitz smooth setting.

**Compliance With Llm Reviewing Policy:**

Affirmed.

**Final Justification:**

The authors addressed all my comments during the rebuttal.

**Key Questions For Authors:**

Most of the theoretical results require Assumptions 1–3, and the saddle-point avoidance
result (Theorem 3) additionally relies on Assumption 5. While these results—especially the
saddle-point avoidance guarantee—are interesting, it is not entirely clear how restrictive
these assumptions are in practical machine learning settings. Could the authors discuss
the applicability of these assumptions and provide examples of real-world learning problems
where they naturally hold? Such discussion would help clarify the practical relevance of the
theoretical guarantees.

**Limitations:**

yes

**Strengths And Weaknesses:**

**Strenghts**
- To the best of my understanding, the theoretical results appear to be technically sound and
are supported by careful analysis. The assumptions adopted in the paper seem appropriate
for the considered setting and are consistent with the scope of the work.
- The paper addresses an important aspect of machine learning optimization, namely the
convergence guarantees. Given the widespread use of learning architectures, extending clas-
sical convergence results to more general settings—such as objectives that are locally but
not globally Lipschitz smooth—is a relevant theoretical direction. In this sense, the work
contributes to improving the understanding of the stability and convergence behavior of
learning algorithms under weaker assumptions.
- The paper extends existing convergence results from the deterministic non-convex setting to
the stochastic case under a taming scheme. While the general idea builds on prior work, the
analysis provides additional insight into the behavior of stochastic gradient methods under
weaker smoothness assumptions.

**Weaknesses**
- While the individual components of the paper—such as the assumptions, theorems, and
claims—are presented clearly, the overall narrative could be improved. In particular, the
connections between sections are not always well articulated, which makes it difficult for the
reader to see how the discussion progresses from one part of the paper to the next and how
the different results fit together within the broader argument.
- The role of the empirical section could also be clarified. For example, Figure 1 is included but
not explicitly discussed in the text, and the motivation for the chosen experimental setting
is not fully explained. It would be helpful for the authors to clarify why this simulation
setup is considered, what specific question it is intended to address, and what conclusions
should be drawn from the results.
- The motivation for relaxing the classical assumptions could be further clarified. The paper
builds on existing convergence results that typically require either bounded iterates (i.e.,
$\sup_t \|x_t\| < \infty$ with probability 1) or global Lipschitz smoothness. However, it is not en-
tirely clear how restrictive these assumptions are in practical machine learning applications.
In particular, the paper could benefit from a discussion on how commonly non-globally
Lipschitz smooth objectives arise in practice and in which settings the proposed analysis
provides a clear advantage.

---

> ### Author Rebuttal · Authors · 2026-03-31
>
> Dear reviewer,
>
> Thank you for your time and positive evaluation! We reply to your remarks and questions below:
>
> > While the individual components of the paper are presented clearly [...] the connections between sections are not always well articulated.
>
> The main idea of our narrative structure was to introduce the various components in a piecemeal fashion that would make them easier to digest. This is why we started with the simplest taming scheme (TSGD) before introducing its dissipative counterpart (DTSGD), which included two distinct mechanisms intended to mitigate the impact of very large stochastic gradients. We will make this narrative structure clearer from the beginning, and if you have any more detailed comments, we will be happy to take them into account in the first revision opportunity.
>
> > Figure 1 is included but not explicitly discussed in the text, and the motivation for the chosen experimental setting is not fully explained
>
> Point taken—we relied on Appendix F to describe the details and motivation, but we understand the back-and-forth can be taxing, so we will transfer all relevant information to the body of the paper. All in all, our experiments have a complementary character on ourtheoretical results and are not meant to be exhaustive. Their goal is, instead, to show that the proposed taming policy can perform at least as well as other methods without theoretical stability guarantees (cSGD, nSGD) and is more stable than vanilla SGD, in a synthetic example of training a NN. The precise framework is chosen so that the objective's gradient grows superlinearly, thus entering in the $(L_0,L_1)$ framework. The performance is measured in terms of stability of the methods with respect to the magnitude of the initialization of the stepsize $\gamma_0$. Stability is then measured as the number of iterations to reach a certain precision by progressively increasing $\gamma_0$, which in practice one may not know how to tune.
>
> > The motivation for relaxing the classical assumptions could be further clarified [...] The paper could benefit from a discussion on how commonly non-globally Lipschitz smooth objectives arise in practice and in which settings the proposed analysis provides a clear advantage.
>
> Consider the following list of examples (see also our reply to [Reviewer 3VPk above](https://openreview.net/forum?id=v7i8l6FXh9&noteId=Oo9NlL267a)):
> - **Deep neural nets** (see also the discussion in L294, right before Theorem 1): in an $L$-layer feed-forward neural network with any of the standard activation functions (ReLU, softplus, leaky ReLU, GELU...), the problem's empirical risk minimization objective grows as $\mathcal{O}(\|x\|^{2L})$, so it is not Lipschitz smooth—but it is $(L_0,L_1)$-smooth.
> - **Matrix factorization problems** (including matrix completion, and robust PCA models): the base objective function here is of the form $f(U ,V) = \|UV − M\|_F^2$. This objective exhibits quartic growth, so it is not Lipschitz smooth—but it is $(L_0,L_1)$-smooth.
> - **Phase retrieval:** The canonical (real) phase retrieval model is of the form $f(x) = \frac{1}{N} \sum_{i=1}^N \big[(a_i^\top x)^2 - y_i\big]^2$. This objective exhibits quartic growth, so it is not Lipschitz smooth—but it is $(L_0,L_1)$-smooth.
>
> All these examples are hallmark machine learning problems (especially in the context of deep learning), and SGD is the go-to method for solving them. However, since the analysis of SGD typically relies on Lipschitz smoothness—and none of the above problems is Lipschitz-smooth—this gives rise to a fundamental tension between the existing theory for SGD and the problems in which it is applied.
>
> Our paper aims to resolve precisely this disconnect. We are happy to elaborate further on these examples and include a dedicated section at the first revision opportunity.
>
> > Most of the theoretical results require Assumptions 1–3 [...] Could the authors discuss the applicability of these assumptions and provide examples of real-world learning problems where they naturally hold? Such discussion would help clarify the practical relevance of the theoretical guarantees.
>
> A standard example would be an empirical risk minimization problem resulting from a deep NN with, say, ReLU activation functions and a regularized quadratic loss. We describe this model in Section 2.3, and we revisit it in L294–300, right before Theorem 1. This problem was the main motivation for our work, so we were particularly careful to make sure that our assumptions are well-calibrated to this model.
>
> That said, we understand that the discussion in L294–300 can be easy to miss, so we will make sure to restructure our presentation to make it more visible at the first revision opportunity.
>
> ---
>
> Thank you again for your time and positive evaluation. We will implement the changes outlined above at the first possible revision opportunity—in the meantime, please let us know if you have any remaining questions.
>
> Kind regards,
>
> The authors

---

> > ### Author Rebuttal · Reviewer_Napj · 2026-04-04
> >
> > I appreciate the authors’ careful rebuttal, which fully addresses all of my comments. Accordingly, I have increased my score.

---

> > > ### Author Response · Authors · 2026-04-05
> > >
> > > Dear Reviewer Napj,
> > >
> > > Thank you for your reply, your positive feedback, and your increased support. We are very happy that our response addressed your concerns.
> > >
> > > Kind regards,
> > >
> > > The authors

---

### Official Review · Reviewer_3VPk · 2026-03-13

**Soundness:** 3
**Presentation:** 3
**Significance:** 3
**Originality:** 3
**Overall Recommendation:** 4
**Confidence:** 1

**Summary:**

This paper considers the convergence analysis of SGD under assumptions which are weaker than Lipschiz smoothness. It shows that by introducing a taming scheme (an adaptive choice of the step-size), almost sure convergence to a critical point can be ensured, assuming L0-L1 smoothness and an additional ABC-type condition. Moreover, by adding an additional dissipation term, a similar result can be obtained assuming only the L0-L1 smoothness.

**Compliance With Llm Reviewing Policy:**

Affirmed.

**Final Justification:**

All remaining questions were adequately addressed

**Key Questions For Authors:**

1. What is the role of $r$ in Theorem 1 and Theorem 2? Choosing $r$ very large does not seem to impact the convergence result. Does it appear only in the rate of convergence?

**Limitations:**

Yes

**Strengths And Weaknesses:**

This paper is very much outside of my area of expertise, and I have not checked the proofs, so I cannot comment on Soundness, Significance and Originality.

Concerning the Presentation of the paper: this is a clearly written and well structured paper, written in a way that is accessible to readers with basic optimization knowledge.

---

> ### Author Rebuttal · Authors · 2026-03-31
>
> Dear reviewer,
>
> Thank you for your time and positive evaluation! We reply to your remarks and questions below:
>
> > What is the role of $r$ in Theorem 1 and Theorem 2? Choosing $r$ very large does not seem to impact the convergence result. Does it appear only in the rate of convergence?
>
> Yes, as the name suggests, the role of the scale function $r(x)$ is to tame the growth of $\|\nabla f(x)\|$ as $\|x\|\to\infty$. As such, the "right" choice for $r(x)$ depends on the class of problems being considered; we provide some concrete guidelines below:
> - **In deep learning** (cf. the discussion in L294, right before Theorem 1): in an $L$-layer feed-forward neural network with any of the standard activation functions (ReLU, GELU, softplus,...), gradients grow as $\|x\|^{2L-1}$, so a reasonable choice would be to take $r(x) = 1 + \|x\|^{2L-1}$.
> - **In matrix factorization** (including matrix sensing, matrix completion, and robust PCA models): here, gradients grow cubically in the problem's control variable (a matrix), so the most principled choice is to take $r(x)=1 + \|x\|^3$.
> - **In phase retrieval:** the classical smooth least-squares phase retrieval objective is quartic in the problem's control variable (in both the real and complex case), so the most principled choice is, again, $r(x) = 1+\|x\|^3$.
> - **In tensor PCA / rank-one approximation problems:** in the standard multilinear / homogeneous formulation with $m$-th order tensors, the objective is an $m$-degree polynomial; with least-squares fitting, the degree is $2m$. As such, a reasonable choice for $r$ would be $r(x) = 1 + \|x\|^{m-1}$ or $r(x) = 1 + \|x\|^{2m-1}$, depending on the loss surrogate.
> - **Black-box:** if the optimizer has no information whatsoever about $f$, the $(L_0,L_1)$-smoothness requirement would advocate taking $r(x)=\exp(x^{3/2})$ (actually, any exponent of the form $x^{1+\epsilon}$ with $\epsilon>0$ would work, the suggestion $\epsilon=1/2$ is only made for concreteness). This case is not very relevant from a practical standpoint, but it is still important as a theoretical worst-case instance (which is why the setting of $r(x)$ is so conservative in this case).
>
> As you point out, this choice only impacts the transient, pre-convergence phase, where large stochastic gradients are most likely to occur. Since the scale factor is weighted itself by the sequence $\lambda_t$ the long-run convergence properties of the proposed taming scheme are not otherwise affected.
>
>
>
> ---
>
> Thank you again for your time and positive evaluation. We will of course be happy to include a version of the above discussion at the first possible revision opportunity—please let us know if you have any further questions in the meantime.
>
> Kind regards,
>
> The authors

---

> > ### Author Rebuttal · Reviewer_3VPk · 2026-04-01
> >
> > Thank you. My concerns have been adequately addressed. This did not help increase my confidence in my ability to assess this work properly, though, so I maintain my score (and my low confidence).

---

> > > ### Author Response · Authors · 2026-04-02
> > >
> > > Thank you for your reply, your time, and your continued positive evaluation, we are very happy that your concerns were adequately addressed.
> > >
> > > Best,
> > >
> > > The authors

---

### Decision · Program_Chairs · 2026-04-30

**Decision:**

Accept (regular)

**Comment:**

This paper proposes a novel taming scheme to study SGD under the generalized smoothness condition and provided convergence guarantees. It is well written, with clear contributions to the optimization theory of SGD for regimes of interest to deep learning.